# USP50 suppresses alternative RecQ helicase use and deleterious DNA2 activity during replication

Hannah L. Mackay[1,12], Helen R. Stone[1,6,12], George E. Ronson [1], Katherine Ellis[1,7], Alexander Lanz[1], Yara Aghabi [1], Alexandra K. Walker[1], Katarzyna Starowicz[1,8], Alexander J. Garvin [1,9], Patrick Van Eijk[2,3], Stefan A. Koestler [1], Elizabeth J. Anthony[1], Ann Liza Piberger[1], Anoop S. Chauhan [1], Poppy Conway-Thomas [1], Alina Vaitsiankova [4,10], Sobana Vijayendran[1,11], James F. Beesley[1], Eva Petermann [1], Eric J. Brown[5], Ruth M. Densham [1], Simon H. Reed [2,3], Felix Dobbs[2,3], Marco Saponaro [1] & Joanna R. Morris [1] ✉

Mammalian DNA replication relies on various DNA helicase and nuclease activities to ensure accurate genetic duplication, but how different helicase and nuclease activities are properly directed remains unclear. Here, we identify the ubiquitin-specific protease, USP50, as a chromatin-associated protein required to promote ongoing replication, fork restart, telomere maintenance, cellular survival following hydroxyurea or pyridostatin treatment, and suppression of DNA breaks near GC-rich sequences. We find that USP50 supports proper WRN-FEN1 localisation at or near stalled replication forks. Nascent DNA in cells lacking USP50 shows increased association of the DNA2 nuclease and RECQL4 and RECQL5 helicases and replication defects in cells lacking USP50, or FEN1 are driven by these proteins. Consequently, suppression of DNA2 or RECQL4/5 improves USP50-depleted cell resistance to agents inducing replicative stress and restores telomere stability. These data define an unexpected regulatory protein that promotes the balance of helicase and nuclease use at ongoing and stalled replication forks.

DNA replication is fundamental for genomic integrity. Obstacles to replication, including unrepaired DNA lesions or extensive secondary structure, can block the progression of replicative polymerases, causing fork stalling, fork collapse, and generating DNA breaks[1]. Hundreds of forks may stall during each S phase in a human cell, and the frequency increases in cells exposed to genotoxic or oncogenic stresses. Pathways to recover stalled and broken replication forks are utilised to resolve impediments to replication so that DNA synthesis can be completed. These pathways include reversal and stabilisation followed by restart; repriming; post-replicative repair; template switching; and double-strand break (DSB)-mediated recovery. Faults in processing

obstacles or restoring replication following processing increase genomic instability, leading to tumorigenesis[2].

RecQ helicases are a highly conserved family of helicases with essential roles in replication and DNA repair[3]. They contain the core helicase domain (DEAD/DEAH box, helicase conserved C-terminal domain) and possess 3' to 5' unwinding directionality capable of unwinding a variety of structures; they can also anneal complementary ssDNA and perform branch migration (reviewed in refs. [4],[5],). There are five human RecQ helicases: RECQL1, WRN, BLM, RECQL4, and RECQL5 (reviewed in ref. [6]). Four are linked to human syndromes characterised by cancer predisposition and/or premature ageing: Werner's syndrome (WRN), Bloom's syndrome (BLM), and Rothmund-Thomson,

RAPADILINO, and Baller-Gerold syndromes (*RECQL4*)[4,7]. Recently, two families with a genome instability disorder, RECON syndrome, have been found to carry biallelic mutations in *RECQL1*[8]. The WRN helicase has been identified as a synthetic lethal target of cancers with high levels of microsatellite instability[9–12]. Its helicase activity is required to process cruciform structures formed of large $(TA)_n$ repeats generated through microsatellite instability over time[13].

Three of the RecQ helicases are known to be employed to restart stalled replication forks. BLM-deficient cells restart poorly after aphidicolin or hydroxyurea (HU) exposure[14]. RECQL1 restores stalled and reversed forks exposed to TOP1 inhibitors and several other replication stress inducers[15,16]. WRN facilitates the progression of replication in normal physiological conditions or after exogenous genotoxic stress[17–19], has been implicated in the recovery of arrested forks[20,21], and contributes to the processing of stalled and reversed forks to promote restart[22]. Additionally, the RECQL5 helicase may be used under certain circumstances[23,24]. How RecQ helicases, which unwind similar DNA structures in vitro[25–28], are deployed at different times and at different structures in cells is not clearly defined.

Multiple DNA nucleases are similarly crucial to replication fork kinetics and responses to replicative stress (reviewed in ref. 29). FEN1 is critical to Okazaki fragment processing and fork restart[30] and MRE11 processes gaps behind replication forks and is also required for restart[31]. The nuclease DNA2 has recently emerged as critical to stalled replication fork processing in conjunction with WRN, where the DNA2- and WRN-dependent mechanism degrades reversed forks to promote restart[22]. DNA2 also promotes ongoing replication[32–35] and biallelic *DNA2* mutations have been identified in patients with Seckel syndrome and primordial dwarfism, conditions associated with under-replication[36,37]. Recently, compound heterozygosity of *DNA2* mutations has been associated with severe growth failure and the clinical characteristics of Rothmund-Thomson syndrome[38], a condition previously linked to *RECQL4*. Thus, a critical question is how nuclease and RecQ helicase relationships are regulated in certain contexts.

Ubiquitin (Ub) is a versatile protein that acts both as a signal for protein turnover and as a component capable of altering protein-protein interactions. Ub modification pathways are a central means to respond to and fine-tune replication fidelity, acting in both the machinery of unperturbed replication and, most prominently, in the supporting pathways that tolerate, repair, or respond to replication difficulties[39]. Ub-interacting proteins carry one or more structurally diverse Ub-binding domains to drive such exchanges[40]. Ub is conjugated to proteins through a three-enzyme cascade, whereas the processing of Ub from proteins acts to restrain the Ub signal. The role of Ub in replication is complex, and many of the pathways it regulates are poorly understood.

Here, we expand our understanding of the relationship between RecQ helicases and nucleases during replication. We find that the Ub-binding face, but not de-ubiquitinating activity of USP50, a member of the ubiquitin-specific-processing protease (USP) family of de-ubiquitinating enzymes, promotes replication kinetics. USP50 is needed to support ongoing replication, suppress ssDNA exposure, promote telomere stability, and prevent MUS81-dependent DSB foci formation. USP50 also supports cellular resistance to the replication stalling agent HU and to the G4-quadruplex stabilising agent pyridostatin.

We discover that USP50 acts to encourage WRN-FEN1 localisation at or near replication forks and over-expression of FEN1 or WRN can overcome the need for USP50 in the promotion of ongoing replication, the promotion of fork restart and the suppression of spontaneous breaks. Moreover, in cells lacking USP50, we find that the DNA2 nuclease and RECQL4 and RECQL5 helicases promote fork stalling, poor recovery, ssDNA exposure and fork collapse. Similarly, in cells depleted for FEN1, fork restart is suppressed by DNA2 and RECQL4/5. Suppression of DNA2 improves HU resistance and telomere stability,

and RECQL4 and RECQL5 suppression improves pyridostatin resistance of USP50-depleted cells. These findings reveal an unexpected regulator of helicase and nuclease use during replication.

## Results

### USP50 Ile-141 promotes ubiquitin binding and chromatin recruitment

A previous RNA interference screen highlighted USP50 as potentially important to replication[41]. The USP class of de-ubiquitinating enzymes are cysteine proteases characterised by a catalytic domain divided into a series of conserved regions. Human USP50 (uniprotkb/Q70EL3) is classified as a non-protease homologue of USPs as it lacks the conserved acidic residues required for the activity of USP domain enzymes and it fails to process Ub-β-galactosidase[42]. The Alphafold USP50:Ub structure predicts a complex similar to structures reported for USP domains with Ub (e.g., PDB: 3n3k[43]). In the examination of USP50 protein sequences from 24 diverse species, we noted 49 invariant and 39 highly conserved amino acids of the 339 total, suggesting the conservation of some aspects of the protein, including its predicted Ub interaction face (Fig. 1A, Supplementary Fig. 1). Many proteins, including the USP-family of de-ubiquitinating enzymes, interact with Ub through Ub's hydrophobic, Leu8, Ile44, and Val70 residues[44]. In the predicted USP50:Ub structure, Ub Ile44 is close to USP50 Ile-141 (Fig. 1A), and leucine or isoleucine in USP50 is at this position in all 24 species (Supplementary Fig. 1). To test whether USP50 can bind Ub we expressed FLAG-USP50 and an I141R-FLAG-USP50 mutant along with Myc-tagged Ub (Myc-Ub) and performed FLAG immunoprecipitation. We found that WT-USP50 co-precipitated high molecular weight Ub conjugates. By comparison, the mutant co-precipitated ~35% of the conjugates precipitated by the WT protein (Fig. 1B), suggesting that USP50 interaction with Ub conjugates is in part through its predicted Ub-binding face. Using purified Ub, and bacterially expressed USP50, we found USP50 bound Ub chains but not free Ub (Supplementary Fig. 2A, B), suggesting a preference for conjugates.

If USP50 has a role in replication, it might be expected to be associated with chromatin. We fractioned cells expressing exogenous FLAG-USP50 and noted a proportion of USP50 co-purified with chromatin, which was increased following HU treatment (Fig. 1C). Intriguingly, I141R-FLAG-USP50 exhibited less chromatin enrichment, and its association with chromatin did not increase following HU treatment (Fig. 1C). Ub conjugates can be directed for degradation by the proteasome or unwound by the p97 AAA+ ATPase segregase. We treated cells with the proteasome inhibitor MG132 to reduce Ub conjugate turnover and the VCP/p97 inhibitor CB-5083 to reduce extraction of Ub-conjugated proteins[45]. Both treatments increased Ub conjugates in whole-cell lysates, although proteasome inhibition enriched Ub conjugates co-purified with chromatin more than VCP inhibition. MG132 treatment, but not VCP suppression, also increased the chromatin association of USP50 (Fig. 1D). This analysis also revealed high molecular weight bands immunoreactive with anti-FLAG following MG132 treatment, suggesting USP50 is itself Ub-modified and processed through the proteasome.

The increased co-purification of USP50 with chromatin observed after HU treatment suggested the potential for association at or near stalled replication forks. To further assess USP50 localisation with bulk DNA in comparison to DNA at stalled forks, we tested USP50 proximity to the nucleotide analogue EdU after its incubation under two different conditions; either for 24 hours to ensure analogue incorporation throughout the DNA or following a short 15-minute pulse followed by 3 hours of HU treatment to label nascent DNA at stalled forks[46]. We used a proximity ligation assay (PLA) with antibodies to the tagged nascent DNA and to the FLAG tag fused to USP50. This method indicated proximity of FLAG-USP50 with both long-label EdU, and short-label EdU/HU treatments (Fig. 1E). The I141R-FLAG-USP50 mutant exhibited less signal in comparison to the WT-USP50 with short-label

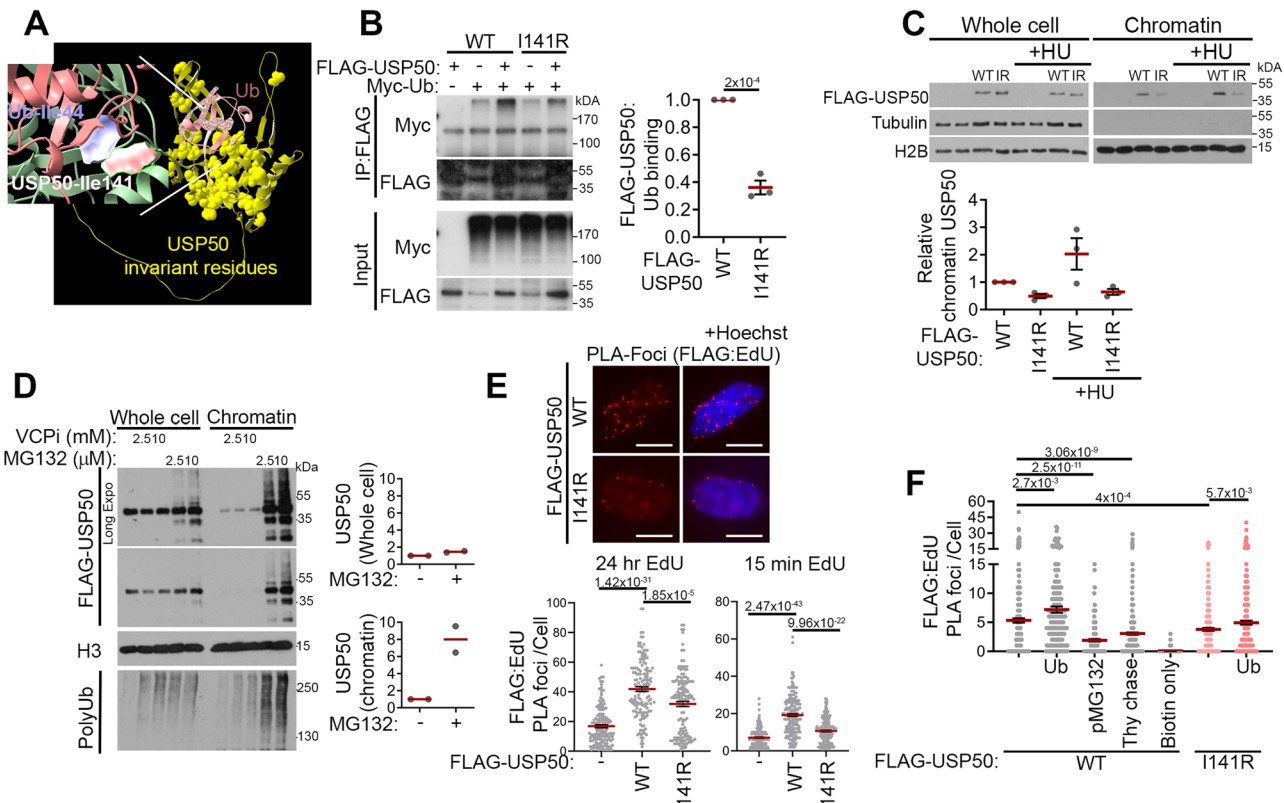

**Fig. 1 | USP50 Ile-141 promotes Ub-binding and chromatin recruitment after HU treatment.** Where included, graphs indicate the mean ± SEM, exact *P* values are shown, and number of biological repeats is listed (*n*). All statistical analysis in this figure was performed using a two-tailed unpaired *t* test. Source data are provided with this paper. **A** USP50:Ub interaction predicted by Alphafold2. Electrostatic densities of invariant USP50 residues are shown (yellow). In the inset Isoleucine-44 of Ub and Isoleucine-141 of USP50 are shown as electrostatic density. **B** Immunoprecipitation of FLAG epitopes from HeLa cells expressing FLAG-USP50 or I141R-FLAG-USP50 and Myc-Ub, probed for FLAG and Myc. Mean Myc-Ub normalised to both Myc-Ub and FLAG-USP50 expression in the whole-cell lysate. *n* = 3. **C** Immunoblot of whole-cell lysate and the chromatin fraction of FLAG-USP50 or I141R-FLAG-USP50, Tubulin and H2B from cells untreated or treated with 5 mM HU for 3 hours. Graph shows mean FLAG-USP50 in the chromatin fraction relative to FLAG-USP50 in the untreated sample. *n* = 3. **D** FLAG-USP50 expressing cells were treated with VCPi (CB-5083) for 3 hours, or MG132 for 4 hours before lysis. Immunoblots show whole-cell lysate and chromatin fraction. FLAG-USP50 expression for NTC vs 10 μM MG132 was quantified and shown for both the whole-cell extract (top right graph) and chromatin-enriched fraction (bottom right). *n* = 2. **E** Mean proximity ligation assay (PLA) foci of FLAG-USP50 or I141R-FLAG-USP50 with Biotin-EdU. DNA was either labelled with EdU for 24 hours (bottom left) or 15 mins followed by 3 hours 5 mM HU treatment (bottom right). Representative images (top) of the 24 hours EdU incubation with 10 μm scale bar. *n* = 3, >150 cells per condition. **F** Mean PLA foci of FLAG-USP50 or I141R-FLAG-USP50 with Biotin-EdU after 5 mins EdU treatment followed by 3 hours 5 mM HU, with, or without, expression of Myc-Ub. Control conditions were pre-treatment with 5 μM MG132 1 hour before EdU treatment (pMG132) or 100 μM thymidine added for 5 mins after EdU treatment, before HU treatment (Thy chase). *n* = 3, >196 cells per condition.

EdU (Fig. 1E). To test the proximity to nascent DNA further, we used a shorter EdU incubation (5 mins) before HU treatment and tested the impact of allowing 10 mins replication run on (thymidine chase) between EdU treatment and before the addition of HU. A proximity signal was evident when EdU incorporation immediately preceded HU treatment, which was reduced when interrupted by 10 mins of thymidine (Fig. 1F), suggesting the interaction is closer with nascent DNA. We also compared the impact of Ub over-expression prior to EdU or HU with the proteasome inhibitor MG132. Suppressing the proteasome increases Ub in conjugates (and increases USP50 interaction with chromatin (Fig. 1D), but also rapidly inhibits free Ub levels, thereby suppressing new conjugate formation[47]. Consistent with Ub-regulated proximity of USP50 to nascent DNA, the signal between FLAG-USP50 and EdU was increased following Ub over-expression and reduced when MG132 was added prior to EdU (Fig. 1F). We also examined the impact of Ub over-expression on I141R-FLAG-USP50: EdU proximity, the mutant showed reduced proximity foci with EdU compared to WT, and the proximity signal was improved slightly by Ub over-expression, indicating that the mutant is not impervious to the impact of Ub. Together, these findings are consistent with the assessment of chromatin association of the WT and I141R mutant proteins with and without HU treatment (Fig. 1C), they suggest the Ile-141 face is particularly significant for association at or near nascent DNA following HU treatment, show USP50 accumulation at or near nascent DNA is Ub-regulated, and that while the I141R mutant has a reduced ability to locate near nascent DNA, it remains in part regulated by Ub.

## USP50 promotes replication in unperturbed and stressed conditions

Human USP50 mRNA is part of cluster 71, defined as a testis-DNA repair cluster (confidence 0.99), and its expression is at low levels in most other tissues[48] (and Human Protein Atlas, proteinatlas.org). Indeed, we could not detect USP50 in HeLa cell lysates by immunoblot. We wished to test whether USP50 protein is relevant to replication, despite its low-level expression, and so generated HeLa cells bearing an isopropyl-β-D-1-thiogalactopyranoside (IPTG) inducible shRNA to USP50 (shUSP50-expressing cells), which we demonstrated depleted wild-type, exogenous FLAG-USP50 (Supplementary Fig. 3A). We next generated a FLAG-tagged USP50 construct resistant to that shRNA, which was integrated into an inducible site, expressed upon doxycycline (Dox) treatment, named FLAG-USP50 hereafter. We made a second line in the same way, except that Ile-141 was mutated to arginine, named I141R-

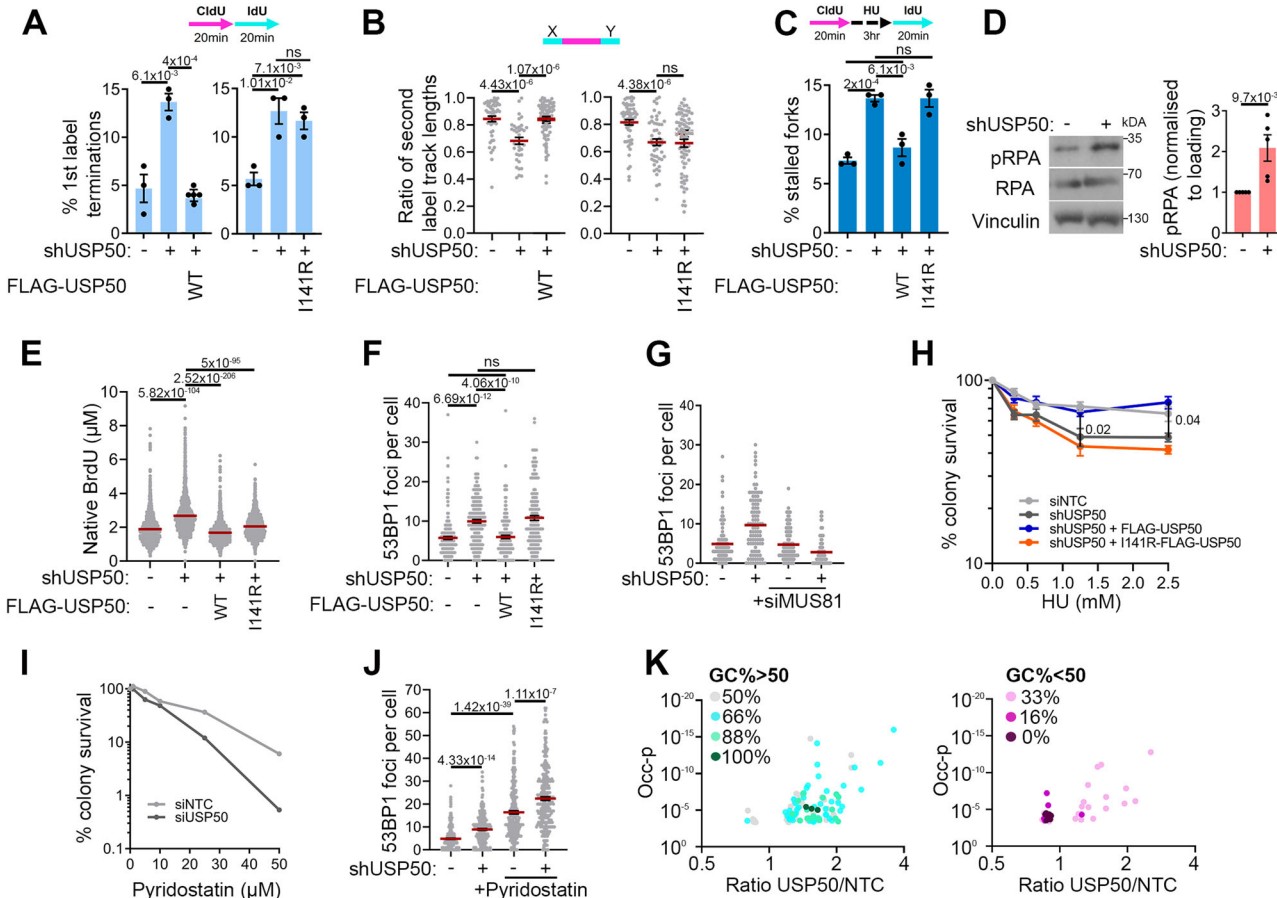

**Fig. 2 | USP50 promotes replication in unperturbed and stressed conditions.** Where included, graphs indicate the mean ± SEM, exact *P* values are shown, and number of biological repeats is listed (*n*). Other than 2H which was two-way ANOVA, all statistical analysis in this figure was performed using a two-tailed unpaired *t* test. Source data are provided with this paper. **A** Mean% of first-label terminations after non-targeting control (NTC) siRNA (−) or shUSP50 (+) and complemented with FLAG-USP50 (left), I141R-FLAG-USP50 (right) or uninduced (−). *n* = 3, >195 fibres per condition. **B** Mean ratio of second-label tracts either side of first label after treatment as in **A**. *n* = 3, >35 first-label origins per condition. **C** Mean% of stalled forks after treatment as in **A**. *n* = 3, >240 fibres per condition. **D** Immunoblot of RPA, pRPA and vinculin following 3 hours 5 mM HU. Quantification shows mean pRPA

from *n* = 5. **E** Native BrdU tracts length after treatment as in **A** and with HU. *n* = 3, >1400 tracks per condition. **F** 53BP1 foci numbers after treatment as in **A**. *n* = 3, >150 cells per condition. **G** 53BP1 foci numbers after treatment as in **A**. *n* = 2, >100 cells per condition. **H** Colony survival after treatment as in A and 16 hours HU. *n* = 4. **I** Colony survival after siNTC or shUSP50 and 24 hours Pyridostatin. *n* = 2. **J** 53BP1 foci numbers after siNTC (−) or shUSP50 (+), with or without 24 hours 100 μM Pyridostatin. *n* = 3, 250 cells per condition. **K** GC content of break-adjacent heximeric sequences enriched or reduced in USP50:siNTC treated cells from overlapping sequences between *n* = 2 biological repeats. Occ-p: *p* value of the occurrence difference. Statistical test: hypergeometric for over/under-representation. All significant sequences and *p* values shown in Supplementary Fig. 4B.

FLAG-USP50 hereafter (Supplementary Fig. 3B). We then examined replication fork structures using the DNA fibre assay, incorporating two nucleotide analogues sequentially and recording the types of structures observed (illustrated in Supplementary Fig. 3C). Strikingly, shUSP50-expressing cells displayed reduced ongoing forks, as indicated by an increase in first-label terminations (CIdU fibres without IdU), and greater asymmetry between second labels from first-label origins, indicative of fork stalling (Fig. 2A, B). In contrast, shUSP50-expressing cells complemented with FLAG-USP50 had replication features comparable to untreated cells, whereas expression of I141R-FLAG-USP50 did not improve replication defects in shUSP50-expressing cells (Fig. 2A, B), suggesting a requirement for the Ile-141 face for ongoing replication.

We examined the stability of forks stalled by HU treatment and observed that shUSP50-expressing cells had slightly shortened nascent DNA, suggesting some reduced ability to protect stalled structures (Supplementary Fig. 3D). We next tested the ability of stalled replication forks to restart, employing an alternative version of the DNA fibre assay in which the second label is applied after washing out the HU. Cells expressing shUSP50 exhibited poor restart, indicated by an elevated level of stalled forks (i.e., those without a second label)

(Fig. 2C). Restart could be improved by FLAG-USP50 expression but not by I141R-FLAG-USP50 mutant expression (Fig. 2C), indicating a requirement for USP50, and the Ile-141 face in the resumption of replication. Perturbations in replication fork progression can lead to replicative helicase-polymerase uncoupling, resulting in the accumulation of single-stranded DNA (ssDNA), which is rapidly coated by Replication Protein A (RPA), and phosphorylated[49]. We observed increased phosphorylated RPA levels in HU-treated cells depleted of USP50 (Fig. 2D), suggesting increased ssDNA. We used the SMART assay[50] to measure the degree of exposed ssDNA directly, growing cells in the presence of the nucleotide analogue BrdU before expression, or not of the shUSP50, and FLAG-USP50 variants. BrdU is detectable with an antibody only when exposed within ssDNA. In shUSP50-expressing cells given a short exposure to HU, we observed increased lengths of exposed BrdU, which could be suppressed by FLAG-USP50 expression but only partially suppressed by I141R-FLAG-USP50 mutant expression (Fig. 2E). Thus, USP50 suppresses the appearance of ssDNA, in part through its Ile-141 face.

Prolonged fork stalling can result in the processing of fork structures and the generation of DNA DSBs[51]. We examined cells for proteins recruited to chromatin around DNA DSBs, 53BP1 and BRCA1.

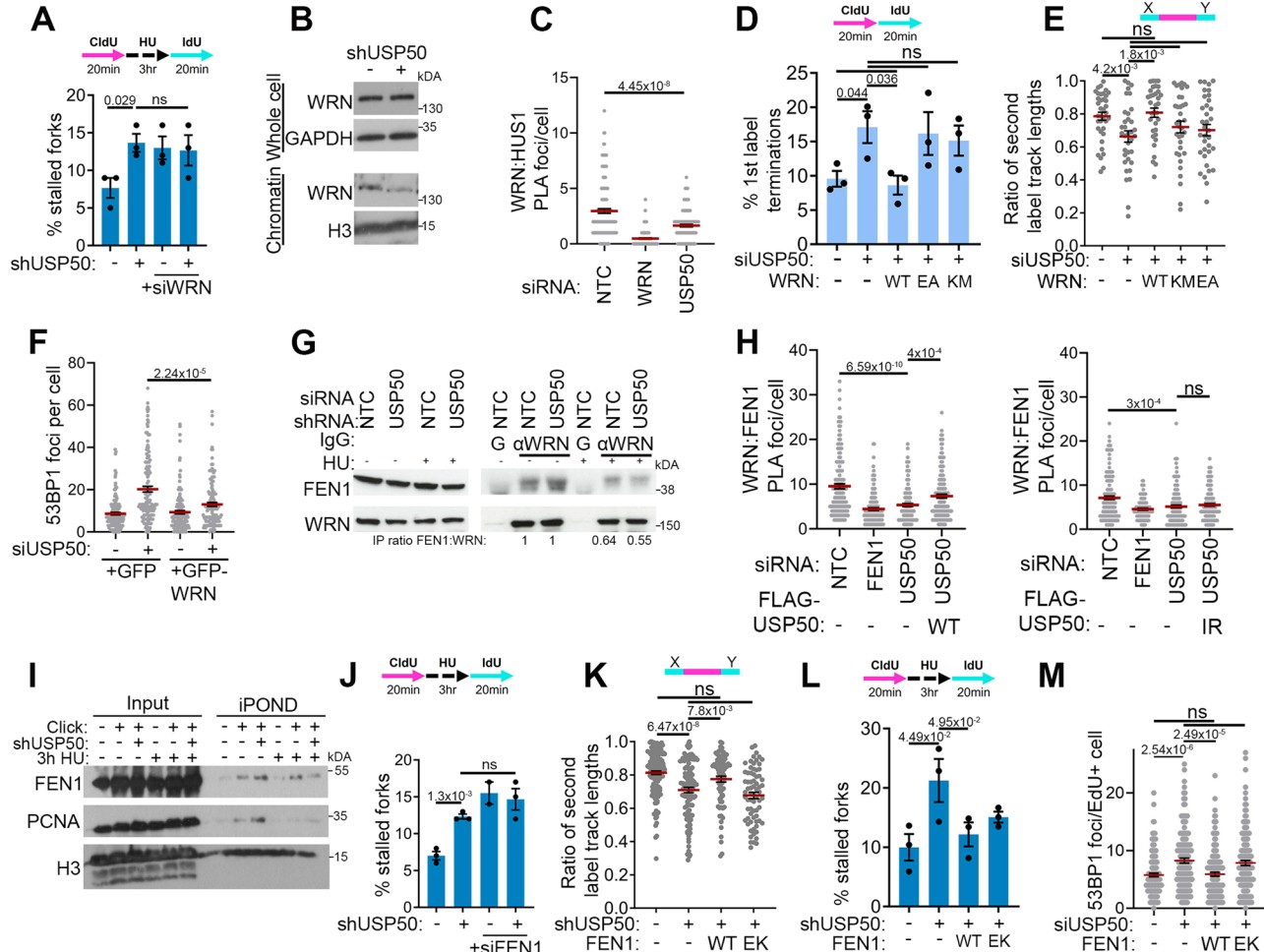

**Fig. 3 | USP50 promotes WRN-FEN1 localisation at stalled forks.** Where included, graphs indicate the mean ± SEM, exact *P* values are shown, and number of biological repeats is listed (*n*). All statistical analysis in this figure was performed using a two-tailed unpaired *t* test. Source data are provided with this paper. **A** Mean% of stalled forks after shUSP50 and WRN siRNA. *n* = 3, >200 fibres per condition. **B** Immunoblot of indicated proteins in whole-cell lysate and chromatin fraction after shUSP50 and 3 hours 5 mM HU (representative of two). **C** Mean proximity ligation assay (PLA) foci of endogenous WRN with HUS1 after siNTC, siWRN, siUSP50 and 3 hours 5 mM HU. *n* = 3, >145 cells per condition. **D** Mean% first-label terminations after siRNA to USP50 and GFP-WT-WRN (WT), GFP-E84A-WRN (EA) or GFP-K577M-WRN (KM) expression. *n* = 3, >200 fibres per condition. **E** Mean ratio of second-label tracts on either side of first labels after treatment as in **D**. *n* = 3, >34 first-label origins per condition. **F** Mean 53BP1 foci after USP50 siRNA and GFP or GFP-WRN expression. *n* = 3, >130 cells per condition. **G** Immunoprecipitation with anti-WRN or control IgG (G) after siNTC, shUSP50 and 3 hours 5 mM HU, probed for WRN or FEN1. Numbers below: ratio of FEN1:WRN relative to siNTC (performed once). **H** Mean PLA foci of endogenous WRN with FEN1 after siNTC, siFEN1, shUSP50, FLAG-USP50 or I141R-FLAG-USP50 expression and 3 hour 5 mM HU. *n* = 3, >150 cells per condition. **I** Immunoblot of indicated proteins input or purified by iPOND after shUSP50 and 3 hours 5 mM HU (representative of three). **J** Mean% stalled forks after shUSP50 and siRNA to FEN1. *n* = 3, >200 fibres per condition. **K** Mean ratio of second-label tracts, either side of first labels after shUSP50 and WT Myc-FEN1 (WT) or E359K-Myc-FEN1 (EK) expression. *n* = 3, >70 first-label origins measured. **L** Mean% of stalled forks after treatment as in **K**. *n* = 3, >200 fibres per condition. **M** Mean 53BP1 foci after treatment as in **K**. *n* = 3, 150 cells per condition.

ShUSP50-expressing cells had increased foci of both proteins (Supplementary Fig. 3E and Fig. 2F). Moreover, 53BP1 foci were suppressed in shUSP50-expressing cells by additional expression of FLAG-USP50 but not by the I141R-FLAG-USP50 mutant (Fig. 2F).

The MUS81 structure-specific endonuclease subunit contributes to the cleavage of persistently stalled replication structures[52–54]. To test whether the 53BP1 foci in shUSP50-expressing cells are because of MUS81, we co-depleted USP50 and MUS81 and counted 53BP1 foci. The combination showed that MUS81 depletion suppressed 53BP1 foci in USP50-depleted cells (Fig. 2G, Supplementary Fig. 3F). Consistent with a replication-associated source of the lesion underlying foci formation, we observed more 53BP1 and BRCA1 foci in cells positive for incorporation of the nucleotide analogue EdU after a short pulse, than those without incorporation (Supplementary Fig. 3E, G). Depletion of MUS81 did not reduce the fork stalling frequency in shUSP50-expressing cells (Supplementary Fig. 3H), suggesting that loss of USP50 acts to stall

forks independently of MUS81. Together, these data correlate poor fork progression, poor restart, and increased ssDNA with increased MUS81-dependent 53BP1 foci, suggesting a proportion of forks stalling because of USP50 loss are subsequently processed to DNA breaks.

Although classified as an inactive de-ubiquitinating enzyme (DUB) (UniProtKB/Swiss-Prot) and unable to process a model substrate[42], some reports have indicated USP50 may have DUB activity (e.g., [55,56].). A recent analysis of several USP-type DUBs suggests variability in the choice of which acidic residue is used as the first or second critical residue (i.e., which aspartic acid)[57]. While human USP50 also lacks the predicted second critical aspartic acid, we nevertheless wished to test for the possibility of activity. We made a further mutant of USP50, substituting both the cysteine and histidine residues critical to USP-DUB catalytic function. The introduction of C53S + H327A (CS-HA) had no impact on the Ub conjugate binding ability of USP50, and when expressed in cells with USP50 depletion, the CS-HA-FLAG-USP50

mutant suppressed the appearance of 53BP1 foci (Supplementary Fig. 3I–K), suggesting USP50 is not acting as a catalytically active DUB in the context of replication stress suppression.

Human *USP50* lies head-to-head with *USP8* on chromosome 15, and the USP50 protein sequence shares 36.6% identity with the C-terminal USP domain of USP8, leading us to consider whether USP8 has a similar function to USP50. Using spontaneous 53BP1 foci to indicate replication difficulties, we compared several siRNA sequences targeting USP50 with those targeting USP8. Exposure of cells to siRNA sequences able to deplete USP50, but not those able to deplete USP8, increased 53BP1 foci in otherwise untreated cells (Supplementary Fig. 3L, M), suggesting USP8 does not share the ability of USP50 to suppress 53BP1 foci generation. We also noted that USP50 siRNA treatment of the human breast cancer cell line MCF7 also increased 53BP1 foci numbers (Supplementary Fig. 3N), as did the treatment of mouse NIH3T3 cells with siRNA targeting murine USP50 (Supplementary Fig. 3N). These data show both human and murine USP50 can act to suppress 53BP1 accumulations, implying a conserved role for USP50 in promoting replication.

To address whether USP50 is relevant to the survival of cells experiencing replicative stress, we examined the ability of shUSP50-expressing cells, complemented with FLAG-USP50, to form colonies after exposure to HU. In comparison to cells treated with siNTC, shUSP50-expressing cells displayed reduced survival following exposure to HU, which was suppressed by complementation with WT-FLAG-USP50 or CS-HA-FLAG-USP50 but not by I141R-FLAG-USP50 (Fig. 2H, Supplementary Fig. 3O). Further, we observed that shUSP50-expressing cells exposed to the G-quadruplex stabilising agent pyridostatin exhibited reduced survival and increased 53BP1 foci formation (Fig. 2I, J). These data indicate that USP50 supports the survival of cells undergoing replicative stress.

Our data indicate that cells without USP50 experience increased fork stalling, a reduced ability to restart stalled forks, increased ssDNA exposure and MUS81-processing into DNA breaks. To understand whether specific regions of DNA are sensitive to USP50 loss, we identified the sequences adjacent to DNA DSB sites in control and USP50-depleted cells by employing INDUCE-seq. The technique uses adaptors fused to DNA DSB ends, allowing sequencing of 300–500 base pairs proximal to the break sites[58]. From 120,000 control siRNA-treated cells, we identified 32,448 break sites, whereas from the same number of USP50 siRNA-treated cells, 147,395 break sites were found. The proportion of break-proximal sequences representing short interspersed nuclear elements, long interspersed nuclear elements, and long terminal repeat elements was lower in USP50 siRNA-treated cells than those treated with control siRNA (Supplementary Fig. 4A), suggesting these regions are not sensitive to USP50 loss. We addressed whether six bp oligonucleotide sequence occurrences at the break-proximal sites differed between USP50 siRNA- versus control siRNA-treated cells, finding both significantly enriched and significantly reduced sequences (Supplementary Fig. 4B). Those sequences enriched near breaks in USP50 siRNA-treated cells had a GC content of 55.6%, whereas sequences significantly enriched near break sites in control cells over USP50-depleted cells had a GC content of just 13.3% (Fig. 2K). The human genome GC content is 40.85%[59]. Thus, these data suggest that USP50 presence suppresses breakage in some GC-rich regions and contributes to breakage in some AT-rich regions.

## USP50 supports FEN1 localisation at stalled forks

To investigate how USP50 influences replication, we tested a possible role for the Werner RecQ helicase, WRN, which contributes both to the replication of GC-rich regions[60,61] and to AT-rich secondary structures[13]. Depletion of WRN in shUSP50-expressing cells did not further increase the percentage of stalled forks observed after HU treatment (Fig. 3A and Supplementary Fig. 5A), suggesting that the two proteins function in the same pathway in fork recovery. We then tested

the possible influence of USP50 on WRN in two ways, assessing HU-treated cells for the amount of WRN co-purified with chromatin and addressing WRN proximity to the RAD9/RAD1/HUS1 (9-1-1) checkpoint clamp, with which WRN interacts following replication stress[21,62]. We observed reduced WRN co-purified with chromatin and a reduced signal of proximity of WRN with the HUS1 component of the 9-1-1 complex following USP50 depletion in HU-treated cells (Fig. 3B, C). These data suggest that USP50 promotes the ability of WRN to associate with chromatin and with the 9-1-1.

We were curious to test whether USP50 impacts WRN turnover and treated cells with the translation-inhibitor cycloheximide to observe WRN degradation rates. We observed no influences of USP50 depletion on either steady-state WRN protein levels or on WRN turnover in untreated cells. Cells treated with HU exhibited a slower rate of WRN loss that was accelerated in cells depleted for USP50 (Supplementary Fig. 5B). When we examined the impact of WT-Flag-USP50 and I141R-USP50 expression, we found that both suppressed WRN loss. Thus, while these data suggest USP50 contributions to the suppression WRN turnover, it is unlikely this function relates directly to the function of USP50 in replication, which requires USP50s Ile-141.

To further test the idea that WRN relates to the defects of USP50-depleted cells, we over-expressed GFP-WRN. Remarkably, GFP-WRN expression suppressed the stalling of ongoing forks, suppressed fork asymmetry and improved restart after HU washout of shUSP50-expressing cells (Fig. 3D, E, Supplementary Fig. 5C, D). Expression of the E84A-WRN mutant that has poor exonuclease function, or the K577M-WRN mutant that perturbs the ATPase/ helicase function of WRN[63], failed to suppress ongoing fork stalling and asymmetry from single origins (Fig. 3D, E). Further, unlike WT-WRN over-expression, these mutant proteins failed to promote fork restart after HU exposure and were unable to suppress the appearance of spontaneous 53BP1 foci of USP50-depleted cells (Supplementary Fig. 5D, E). Thus, WT-WRN over-expression, but not WRN with poor exonuclease or helicase function, can overcome the need for USP50 in promoting ongoing replication, replication recovery after HU treatment, and suppression of fork collapse.

During replicative stress, WRN interacts with nucleases to support replication[22,64–69]. We assessed the interaction of WRN with the flap endonuclease FEN1 and noted a slight reduction in FEN1 co-purified with WRN in HU and shUSP50-expressing cells (Fig. 3G). We assessed the amount of FEN1-WRN in proximity with one another, similarly finding a reduced proximity in USP50-depleted cells (Fig. 3H). The proximity signal between FEN1 and WRN was improved by complementation with FLAG-USP50 but not by I141R-USP50 (Fig. 3H). To further address whether USP50 impacts FEN1 or WRN at replication forks, we used the isolation of proteins on nascent DNA, 'iPOND' method. In the method, newly replicated DNA is purified through a short incorporation of EdU and cycloaddition reaction to tether biotin to the DNA to allow purification of DNA-protein fragments, which are cross-linked through formaldehyde fixation and then the associated proteins analysed[70–72]. WRN peptides have shown enrichment in IPOND samples[72], but as previously described[73,74], we were unable to obtain a signal for WRN in these assays utilising specific WRN antibodies despite testing variants of IPOND designed to improve the detection of larger proteins (RIPA buffer and diluted SDS[74] and native[75]). In contrast, FEN1 and PCNA were detected in control iPOND samples (Fig. 3I). We saw that PCNA association with nascent DNA was reduced following HU treatment as previously reported[72]. FEN1 was retained less on nascent DNA in shUSP50-expressing cells after HU treatment than in control samples (Fig. 3I). Thus, USP50 promotes FEN1 presence at stalled forks. To probe the functional relationship between FEN1 and USP50, we depleted FEN1 in shUSP50-expressing cells. We found that co-depletion did not further increase the percentage of stalled forks observed after HU treatment (Fig. 3J and

Supplementary Fig. 5F), suggesting that FEN1 and USP50 also function in the same pathway in fork recovery.

To further test the role of FEN1 and the WRN-FEN1 interaction, we examined whether FEN1 over-expression might also overcome the need for USP50 and what impact the E359K-FEN1 mutant might have. The E359K mutation abolishes the FEN1-WRN interaction and inhibits the gap endonuclease (GEN) activity of FEN1[76]. In the examination of ongoing forks, we found that WT Myc-FEN1 expression restored ongoing forks measured from a single origin in cells depleted of USP50, but Myc-E359K-FEN1 mutant expression failed to restore fork symmetry (Fig. 3K, Supplementary Fig. 5G). WT Myc-FEN1 expression also restored replication restart after HU exposure, whereas the mutant had an intermediate effect (Fig. 3L). Over-expression of Myc-FEN1 suppressed 53BP1 foci in shUSP50-expressing cells, but the Myc-E359K-FEN1 mutant was unable to suppress the appearance of 53BP1 foci (Fig. 3M), correlating with its impact on ongoing replication. These data suggest WRN interaction-competent and GEN active FEN1 can overcome the need for USP50 in suppressing fork stalling of ongoing forks, but the WRN-FEN1 interaction and GEN activity is partially dispensable for the ability of FEN1 to overcome the need for USP50 depletion at restart of stalled forks. Independently of WRN, FEN1 has a critical role in Okazaki fragment maturation, and cells without FEN1 activity use poly(ADP-ribose) to recruit XRCC1 in a back-up maturation pathway, which can be observed on PARG inhibition[77]. We observed no increased poly(ADP-ribose) in shUSP50-expressing cells (Supplementary Fig. 3G–J), suggesting Okazaki fragment maturation is unaffected by USP50 loss.

Cancer cells bearing high levels of microsatellite instability (MSI-H) use WRN to replicate expanded $(TA)_n$ repeats and are sensitive to loss of WRN[9–11,13]. We wondered whether USP50 contributes to this activity and tested MSI-H colon cancer cell lines, HCT116 and RKO, for sensitivity to USP50 siRNA. However, while these cell lines were susceptible to siRNA targeting of WRN, we found that USP50 siRNA had a minimal impact (Supplementary Fig. 5K). These data suggest no role for USP50 in regulating the functions of WRN associated with supporting cells with high levels of microsatellite instability.

WRN and FEN1 support the replication of the telomeric regions of chromosomes and are particularly implicated in lagging-strand telomeric repeat replication and stability $(TTAGGG)_6$[78–80]. To examine if USP50 has a role in telomere stability, we grew HeLa cells for several days in telomerase inhibitor, with or without shUSP50 expression, complemented or not with FLAG-USP50 constructs. Metaphase spreads were then subjected to chromosome fluorescence in situ hybridisation (FISH) for telomeres labelled by the C-rich probe, $(CCCTAA)_n$, labelling the lagging telomere. We assessed the proportion of chromatids with and without telomeres, observing more chromatids without telomeric stain in cells depleted for USP50 (Supplementary Fig. 5L). Further, FLAG-USP50 expression improved the number of telomeres, whereas complementation with I141R-USP50 did not (Supplementary Fig. 5L). These data suggest that USP50 supports lagging-strand telomere stability.

### Replication defects in USP50-deficient cells are driven by DNA2 and RECQL4/5

In addition to FEN1, WRN also interacts with the nuclease-helicase DNA2[22,81,82], and WRN-DNA2 are together implicated in the restart of reversed forks[22]. We anticipated DNA2 would show reduced localisation to restarting forks following USP50 depletion. However, when we examined DNA2 following USP50 depletion as cells recovered from HU treatment, we observed increased DNA2 foci and an increase in PLA foci between DNA2 and labelled nascent DNA (Fig. 4A, B) but no impact on DNA2 protein levels (Supplementary Fig. 6A). We were intrigued to test whether the increased DNA2 recruitment observed might relate to the defects in fork kinetics observed in USP50-depleted cells. Remarkably, the depletion of DNA2 suppressed ongoing fork stalling, reduced fork asymmetry and improved fork restart in shUSP50-expressing cells (Fig. 4C, D and Supplementary Fig. 6B, C). DNA2 siRNA treatment also suppressed the appearance of increased phosphorylated RPA in shUSP50-expressing cells (Fig. 4E). The selective DNA2 nuclease inhibitor C5 inhibits DNA binding and nuclease activity of DNA2[83]. We tested the inhibitor and found that treating shUSP50-expressing cells with C5 also improved fork restart (Fig. 4F), implicating the nuclease activity in suppressing restart. In contrast, MRE11 inhibition did not improve fork restart (Supplementary Fig. 6D), suggesting the MRE11 exonuclease does not suppress restart in USP50-depleted cells.

Next, we assessed the impact of DNA2 depletion on spontaneous 53BP1 foci as a surrogate measure of collapsed replication forks. While 53BP1 foci were not increased or decreased following depletion of the DNA2 alone, the elevated 53BP1 foci observed in shUSP50-expressing cells were suppressed by co-depletion of DNA2 (Fig. 4G). DNA2 has been implicated in telomere stability[84,85] and as expected its depletion increased lagging-strand telomere loss (Fig. 4H). Remarkably, co-depletion of DNA2 and USP50 increased the proportion of chromatids with telomeres compared to individual depletions (Fig. 4H), suggesting DNA2 is responsible for telomere loss when USP50 is depleted and that USP50 is deleterious to telomere retention in cells depleted for DNA2. Significantly, the depletion of DNA2 also improved the HU resistance of shUSP50-expressing cells (Fig. 4I). These data suggest that excessive or inappropriate DNA2 nuclease activity is responsible for many of the replication defects observed in cells lacking USP50.

The DNA2 nuclease requires the activity of a companion helicase such as WRN or BLM to process DNA around DSBs[22,68,69,81,82,86]. Considering that in USP50-depleted cells, the association of WRN with stalled forks is diminished, we wondered whether an alternative helicase contributes to the replication defects of these cells. We first tested the depletion of helicases BLM and RECQL1, two RecQ helicases important to restart[15,87], finding, as anticipated, that their co-depletion did not suppress the high level of stalled replication structures observed in USP50-depleted cells after release from HU (Supplementary Fig. 7A, B), suggesting neither are responsible for poor fork recovery of shUSP50-expressing cells. In contrast, co-depletion of either RECQL4 or RECQL5 with shUSP50 expression reduced the number of stalled forks after release from HU treatment (Fig. 5A, B and Supplementary Fig. 7C, D). Similarly, we found that co-depletion of either RECQL4 or RECQL5 suppressed ongoing fork stalling of cells treated with USP50 siRNA (Fig. 5C), and the depletion of DNA2, or RECQL4 + 5 suppressed the appearance of long lengths of exposed ssDNA observed following shUSP50 expression and HU (Fig. 5D). RECQL5 depletion alone caused increased 1st label terminations in ongoing replication (Fig. 5C) consistent with a role in ongoing replication in control cells previously described[88–90]. Intriguingly, the slowing/stalling of forks in RECQL5-depleted cells was restored by loss of USP50, suggesting USP50 presence is harmful to ongoing replication when RECQL5 is reduced. We assessed the impact of the helicases on spontaneous 53BP1 foci in USP50-depleted cells, and consistent with the suppression of fork stalling, improved restart, and reduced ssDNA formation seen following RECQL4/5 depletion in a USP50-depleted setting; we found that the elevated 53BP1 foci were suppressed by co-depletion of RECQL4 and RECQL5 (Fig. 5E).

Given our findings that USP50 supports WRN-FEN1 interactions and FEN1 localisation at stalled replication forks (Fig. 3G–I), we wondered if DNA2 and RECQL4/5 might contribute to poor fork recovery in cells lacking FEN1. Indeed, we found that depletion of DNA2 or RECQL4 and RECQL5 also improved fork restart in cells depleted of FEN1 (Fig. 5F). These observations suggest a model in which aberrant use of DNA2 and alternative helicases disrupt fork restart when USP50 is depleted due to reduced FEN1.

We were eager to address the association of RECQL4 and RECQL5 with stalled and recovering forks, however, as for WRN, we could not detect these proteins using iPOND approaches. We, therefore, used

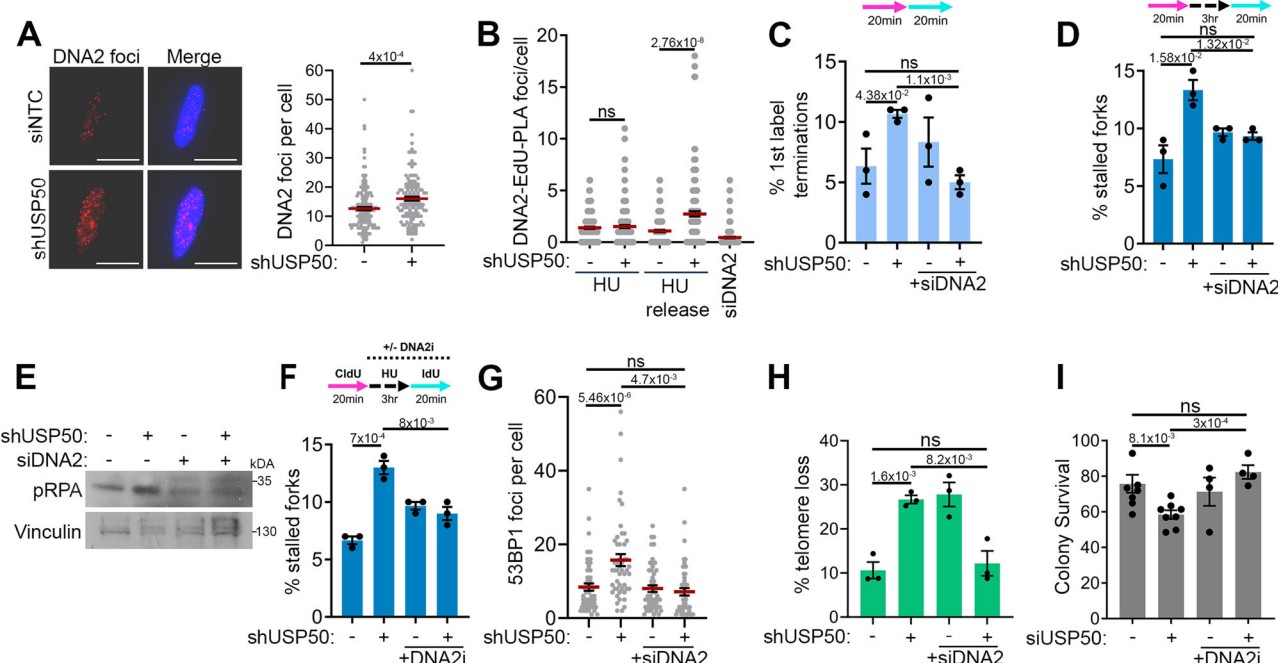

**Fig. 4 | Replication defects in USP50 deficient cells are driven by DNA2.** Where included, graphs indicate the mean ± SEM, exact *P* values are shown, and number of biological repeats is listed (*n*). All statistical analysis in this figure was performed using a two-tailed unpaired *t* test. Source data are provided with this paper. **A** Mean DNA2 foci after shUSP50, 3 hours 5 mM HU, wash and recovery for 20 mins. Representative images (left) with 10 μm scale bar shown. *n* = 3, >140 cells per condition. **B** PLA foci of DNA2 and Biotin-EdU after siNTC (−), DNA2 siRNA, or shUSP50 (+) in cells pulsed with EdU for 15 mins and treated with 3 hours 5 mM HU, or HU-treated and recovered for 20 mins "released". *n* = 3, >150 cells per condition. **C** Mean% of stalled forks after shUSP50, with or without DNA2 siRNA. *n* = 3, >350 fibres per condition. **D** Mean% of stalled forks treated as in **C**. *n* = 3, >200 fibres per condition. **E** Immunoblot of pRPA (S4/8) and Vinculin after shUSP50, with and without DNA2 siRNA and 3 hours 5 mM HU (performed twice). **F** Mean% of stalled forks after shUSP50 with and C5 DNA2i (20 μM). *n* = 3, >200 fibres per condition. **G** 53BP1 foci after shUSP50 with and without DNA2 siRNA and 3 hours 5 mM HU for 3 hours washed and recovered for 20 mins. *n* = 3, >90 cells per condition. **H** Mean% of chromatids with lagging-strand telomere loss after shUSP50 and DNA2 siRNA. *n* = 3, scored chromatid numbers per experiment: NTC: 1482, 956, 898; shUSP50: 1446, 652, 970; siDNA2: 1238, 1236, 714; shUSP50/siDNA2: 170, 1350, 286. **I** Colony survival after USP50 siRNA and 16 hours 1.25 mM HU treatment with or without 20 μM DNA2i. *n* = 6 for siNTC and siUSP40, *n* = 4 for conditions including DNA2i treatment.

indirect immunofluorescence and PLA with labelled nascent DNA. USP50 depletion itself had no impact on RECQL4 or RECQL5 expression (Supplementary Fig. 7E). Intriguingly, using immunofluorescence, we noted that the RECQL4 protein showed increased foci formation in shUSP50-expressing cells recovering from HU exposure (Fig. 5G). On testing the proximity between RECQL4 and nascent DNA in siNTC cells, we observed an increase in proximity signal after release from HU treatment that was further increased in shUSP50-expressing cells (Fig. 5H). While in immunofluorescence assays we did not observe RECQL5 foci, the PLA signal between RECQL5 antibodies and EdU-Biotin was dependent on RECQL5 (Fig. 5I). In contrast to RECQL4, the proximity signal between EdU-Biotin and RECQL5 in control cells was reduced following washout of HU and recovery (Fig. 5I), suggesting RECQL5 at or near recovering forks is typically reduced compared to stalled forks. In USP50-depleted cells, the degree of PLA signal between RECQL5 and EdU-Biotin on HU treatment was less than in control-treated cells; moreover, the signal did not diminish following HU washout (Fig. 5I), suggesting that in USP50-depleted cells a greater degree of RECQL5 is retained at or near forks after HU removal. These findings indicate that USP50 contributes to the correct associations of the RECQL4 and RECQL5 helicases with DNA at or near forks, in particular as they recover from HU removal where USP50 loss is associated with increased RECQL4 association and a failure to diminish RECQL5 association.

Next, we tested the potential of RECQL4 and RECQL5 to influence cell survival of HU- and pyridostatin-treated shUSP50-expressing cells and their influence on telomere stability. We found that the depletion of RECQL4 and RECQL5 reduced the survival of cells exposed to HU,

and their co-depletion with USP50 did not improve resistance to HU (Supplementary Fig. 7F). Similarly, depletion of the RECQL4/5 helicases increased the number of chromatids lacking an associated telomere, and their co-depletion with USP50 did not suppress telomere loss (Supplementary Fig. 7G). These findings are consistent with the previously described roles of RECQL4/5 in replication and telomere stability[89,91–94] and suggest any ability to suppress defects observed on USP50 loss, are insufficient to improve survival after HU exposure or to suppress telomere loss. RECQL4/5 and DNA2 have also been implicated in sensitivity to pyridostatin sensitivity[95]. We found that DNA2 or RECQL4/5 depletion, like USP50 loss, reduced cell survival (Fig. 5J). The co-depletion of DNA2 with USP50 had no impact on survival over those depleted for each alone. Intriguingly, the survival of cells with co-depletion of RECQL4/5 and USP50 showed some suggestion of improved viability over cells depleted for USP50 or RECQL4/5, although not significant (Fig. 5J).

In summary, in USP50-depleted cells, RECQL4, RECQL5 and DNA2 show aberrant localisation, suppress ongoing replication and promote spontaneous 53BP1 foci, they also suppress restart and increase ssDNA following exposure to HU. Depletion of DNA2 improves HU resistance and telomere stability. Thus, RECQL4/5, and particularly DNA2, are responsible for many of the replication defects of USP50-depleted cells.

## Discussion
Cells express multiple helicases and nucleases with critical and complex roles supporting replication. Here, we reveal the surprising finding that the lowly-expressed protein, USP50, promotes the correct

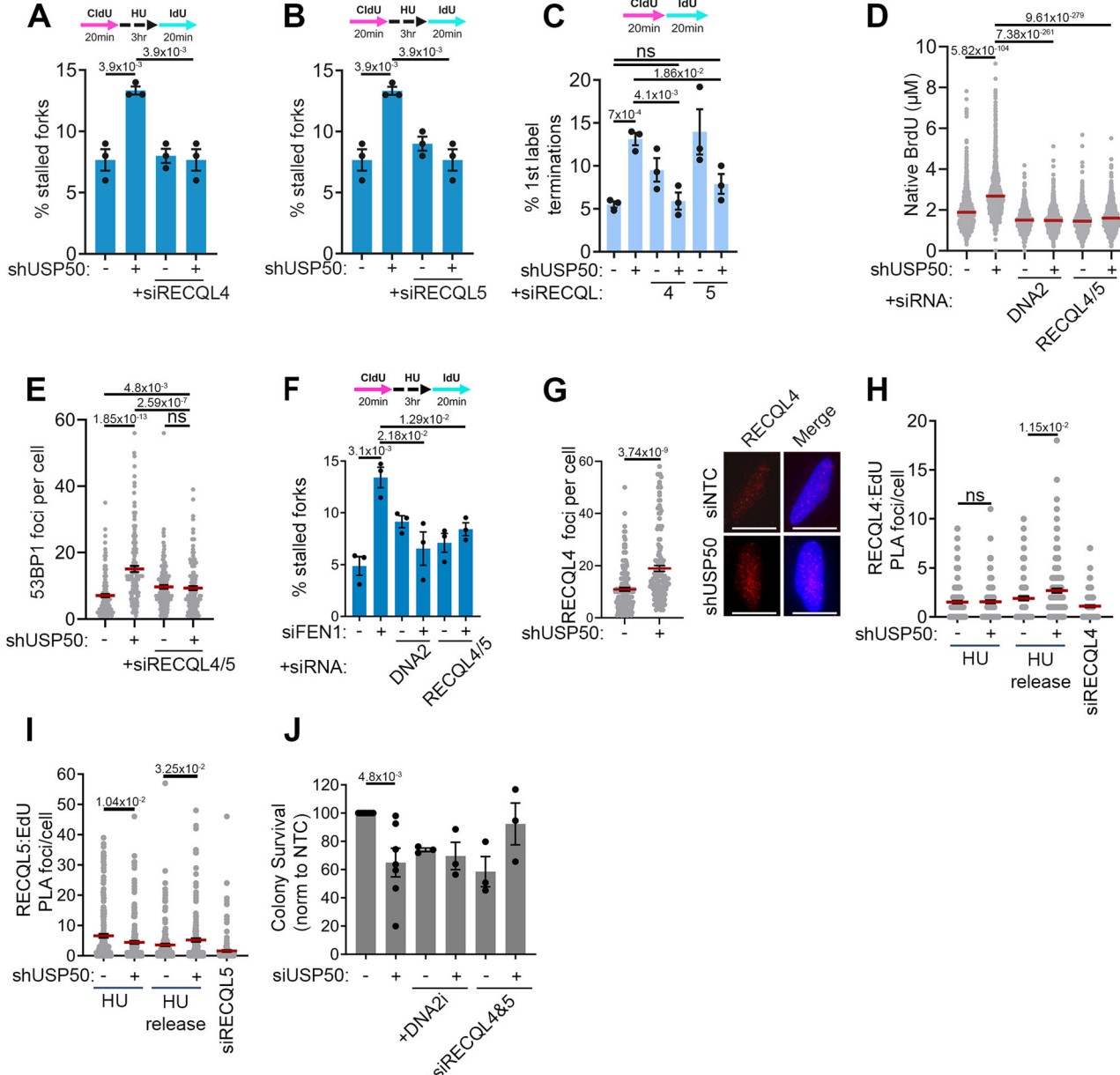

**Fig. 5 | Replication defects in USP50 deficient cells are driven by RECQL4/5.** Where included, graphs indicate the mean ± SEM, exact *P* values are shown, and number of biological repeats is listed (*n*). All statistical analysis in this figure was performed using a two-tailed unpaired *t* test. Source data are provided with this paper. **A** Mean% of stalled forks after shUSP50 with RECQL4 siRNA. *n* = 3, >200 fibres per condition. **B** Mean% of stalled forks after shUSP50 with RECQL5 siRNA. siNTC and shUSP50 conditions are shared with 5 A. *n* = 3, >200 fibres per condition. **C** Mean% of first-label terminations after shUSP50, with or without RECQL4 or RECQL5 siRNA. *n* = 3, >200 fibres per condition. **D** Native BrdU tracts length after shUSP50 and siRNA targeting RECQL4 and RECQL5 or DNA2 and 3 hours 5 mM HU. siNTC and shUSP50 conditions are shared with 2E. *n* = 3, >1400 tracks per condition. **E** Mean 53BP1 foci after shUSP50 and RECQL4 and RECQL5 siRNA and 3 hours 5 mM HU followed by 20 mins recovery. *n* = 3, 150 cells per condition. **F** Mean% of stalled forks after FEN1 siRNA with and without siRNA targeting RECQL4 and RECQL5. *n* = 3, >400 fibres per condition. **G** Mean RECQL4 foci after shUSP50 and 3 hours 5 mM HU, followed by 20 mins recovery. Representative images left, scale bar is 10 μm. *n* = 3, 150 cells per condition. **H** Mean PLA foci of RECQL4 with Biotin-EdU after RECQL4 siRNA, or shUSP50 in cells pulsed with EdU for 15 mins and 3 hours 5 mM HU, or HU treated, washed and allowed to recover for 20 mins (HU release). *n* = 3, >150 cells per condition. **I** Mean PLA foci of RECQL4 with Biotin-EdU after RECQL5 siRNA, or shUSP50 in cells pulsed with EdU for 15 mins and 3 hours 5 mM HU, or HU treated, washed and allowed to recover for 20 mins (HU release). *n* = 4, >200 cells per condition. **J** Colony survival after shUSP50 and siRNAs to DNA2 or RECQL4 and RECQL5 and 24 hours 25 μM Pyridostatin. *n* = 7 for siNTC and shUSP50, *n* = 3 for all other conditions.

balance of nucleases and helicases used during ongoing replication and restart after HU-mediated stalling, and show this promotion is relevant to the survival of cells exposed to HU and pryidostatin and for telomere retention.

Mice homozygous for disrupted *Usp50* (*Usp50^tm1(KOMP)Vlcg*) die in utero[96], indicating it has critical functions. USP50 has been implicated in inflammasome signalling, erythropoiesis, the G2/M checkpoint, and Human Growth-Factor-dependent cell scattering[55,56,97,98]. A previous

siRNA screen identified USP50 as a candidate replication-related protein[41].

We find USP50 at chromatin and near nascent DNA. One hypothesis consistent with our observations is that conjugate(s) at or near the replisome contribute to USP50 retention with an additional contribution of proteasome regulation of USP50 turnover. However, we do not discount the possibility of an alternative, non-Ub interactor, at the Ile-141 face nor a contribution of Ub outside of this face. The

finding that USP50-I141R can suppress WRN turnover in HU-treated cells, points to an activity independent of this face and chromatin co-purification of the I141R mutant is reduced rather than lost, suggesting the contribution of a further feature or factor. Approximately 10% of known mammalian de-ubiquitinating enzymes are predicted to be inactive, 'pseudo-enzymes'. The USP class of pseudo-enzymes with known cellular activities have functions attributed to domains other than their USPs[99]. The current study demonstrates that the catalytically inactive USP region is critical to function.

USP50 loss diminishes, rather than eliminates, measures of WRN-FEN1 at nascent DNA, and over-expression of WRN or FEN1 can overcome the need for USP50. While we were able to examine a direct impact of USP50 loss on FEN1 binding to nascent DNA, our inability to directly detect WRN biochemically must slightly diminish the certainty of our WRN conclusions. USP50 is not related to FEN1- mediated Okazaki fragment maturation, but FEN1 and USP50 are epistatic in supporting ongoing and stalling forks, and the WRN interaction and GEN activity of FEN1 is needed to overcome the requirement for USP50 in ongoing replication and to suppress 53BP1 foci formation. FEN1's GEN activity contributes to the resolution of secondary DNA structures[100] and is associated with break-induced recombination[30,101]. Unlike WRN depletion, USP50 depletion is not lethal to MSI-H cells, suggesting USP50 does not relate to the ability of the WRN helicase to process the non-B form $(TA)_n$ DNA repeats[13]. Indeed, a decrease in the proportion of DNA DSBs near AT-rich sequences and towards GC-rich sequences was observed in USP50-depleted cells. Both WRN helicase and nuclease activities were required to overcome the need for USP50 to promote ongoing forks and suppress fork collapse. WRNs exonuclease can degrade DNA substrates with secondary structures[102,103], and in cells, it may contribute to translesion synthesis replication[104,105]. Its helicase activity can unwind various secondary structures[106,107], and in cells, in addition to processing non-B form $(TA)_n$ DNA[13], the helicase activity is associated with processing unprotected replication forks[108], and restarting of reversed forks[22]. Like WRN and FEN1[76,78,79], USP50 promotes the stability of lagging, G-rich telomeres. USP50-depleted cells are sensitive to the G4-quadruplex stabilising agent, pryidostatin, and form more 53BP1 foci in response, suggesting USP50 may help promote replication through G4Q sites. However, in otherwise untreated cells, G4-quadruplex mapped sequences[109-111] do not represent an increased proportion of the break sequences that occur in USP50-depleted Vs control cells (1.79% Vs 5.3% respectively), suggesting these sequences, of themselves, are not especially vulnerable to the loss of USP50. The mechanism of USP50 support is unlikely to be stoichiometric, given its low expression levels, and whether it contributes to a permissive environment or specifically regulates a particular protein remains to be investigated. We speculate it may regulate an enzyme and, in this way, amplify its influence.

In addition to reduced WRN-FEN1 association with nascent DNA, cells lacking USP50 show an increased association of DNA2 and RECQL4/5 helicases, particularly as forks restart after HU removal, although, again, these conclusions arise from examination through immunofluorescence and proximity to nascent DNA rather than direct biochemical observations. Unexpectedly, but consistent with the aberrant localisation of DNA2, RECQL4 and RECQL5, we find that many of the replication defects in cells lacking USP50 are mediated by the DNA2 nuclease and by RECQL4/5. Further, since cells lacking FEN1 also show DNA2 and RECQL4/5-dependent suppression of replication restart, we suggest a model where USP50's support of FEN1 restricts DNA2 and RECQL4/5 usage (Fig. 6A). Depletion of DNA2 and or RECQL4/5 reduces ssDNA in USP50-depleted cells, suggesting the DNA2 nuclease and RECQL4/5 helicases act together to increase resection at replication forks, which may, in turn, create a substrate for MUS81, resulting in fork collapse and restricting restart (Fig. 6B).

DNA2 usually acts with WRN to promote reversed fork restart and acts to suppress recombination-dependent replication at replication

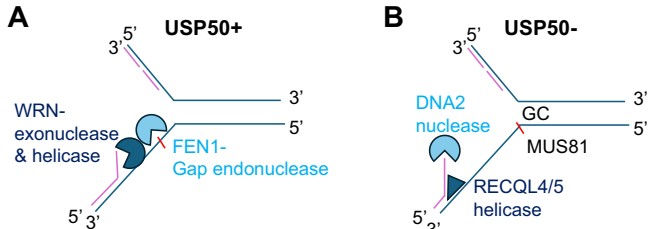

**Fig. 6 | Model of the influence of USP50 on WRN-FEN1, RECQL4/5, and DNA2 during replication. A** USP50 promotes the recruitment of WRN-FEN1 to stalled replication structures. Helicase and nuclease competent WRN (dark blue), and GEN and WRN interaction-competent FEN1 (light blue) can rescue the lack of USP50 to promote ongoing replication and suppress DSB foci. **B** Without USP50, increased RECQL4 and RECQL5 (dark blue) and DNA2 (light blue) result in extended resection and MUS81-dependent DNA breaks. DNA DSBs are more common near CG-rich sequences.

forks[22,112]. Nevertheless, in several contexts, the cell must defend DNA structures from an over-active DNA2 nuclease, for example, to prevent replication fork instability[67-69] or to suppress the expansion of post-replicative gaps[113]. Our data reveals ongoing and restarting forks and telomeres can be similarly vulnerable to excessive DNA2 activity.

RECQL5 can support replication in specific circumstances[23,24] and RECQL4 is a component of the MCM replicative helicase complex and plays a role in replication initiation[114-116], is required for telomere maintenance[91] and has recently been implicated in post-replicative repair[117]. Similar to our observations but in the context of DNA DSBs, the RECQL4 helicase promotes DNA end resection[118] and RECQL5 is reported to be retained at DNA damage sites longer in WRN-deficient cells[24]. We find that reducing RECQL5 or RECQL4 is sufficient to promote ongoing replication and restart after HU release in USP50-depleted cells, implying a need for both enzymes in the suppression of replication and fork restart. Their precise contribution requires more assessment. For example, they poorly destabilise ssDNA secondary structures (in contrast to WRN or BLM) and exhibit relatively strong strand annealing[82,119-121]. Our data suggest that alternative helicase engagement can suppress replication.

Our data suggests that the balance between USP50-mediated functions and the RECQL4/5 and DNA2 axis will likely depend on context. For example, over and above the deleterious impact of individual depletions of USP50, DNA2 or RECQL4/5, the depletion of both DNA2 and USP50 improves telomere stability, implying that an excess of either axis is harmful under certain conditions.

In summary, we have identified USP50 as a regulatory factor influencing nucleases FEN1, DNA2, and helicases WRN, RECQL4 and RECQL5 at ongoing and stalled replication forks, having an impact on cell survival in response to replicative stressing agents and telomere stability.

## Methods
### Tissue culture
HeLa, MCF7 and NIH3T3 (ATCC, CRL-1658) cells were grown in Dulbecco's Modified Eagle Media supplemented with 10% FBS and 1% PS. HCT116 were grown in McCoy's 5 A medium with 10% fetal bovine serum (FBS) + 1% penicillin/streptomycin (PS). RKO cells were grown in Minimum Essential Medium with 10% FBS, 2 mM L-glutamine and 1% PS. Cells were cultured in Corning T75 flasks and 10 $cm^2$ plates and kept at 5% $CO_2$ and 37 °C. Details of key chemicals are in Supplementary Table 3. FlpIn HeLa (human female), MCF7 (human female) and NIH3T3 (murine male) cells were from Morris stocks (from commercial or colleague sources). HCT116 (human male) and RKO (human unspecified sex) were a gift from Prof Andrew Beggs, University of Birmingham.

## Plasmid/siRNA transfection and inducible shRNA

Custom Lentiviral shUSP50 sequence (based on the USP50-7 siRNA sequence C UAC CCA GCA UUU ACG) or NTC sequence cloned into the pLKO-puro-IPTG-3xLacO vector were made by Sigma-Aldrich (Merck). Flp-In™ HeLa cells were lentivirally infected with NTC or USP50 shRNA as per the manufacturers' protocol and then cells selected using Puromycin. Clones were tested for the ability to knock down expression of FLAG-USP50 after 100 μM IPTG for 72 hours and to phenotypically increase spontaneous 53BP1 foci formation following treatment with shUSP50. FuGENE 6 (Roche) was used as a reagent to transfect cells with DNA plasmids. The ratio used was 4:1 FuGENE (μl:μg DNA), following the manufacturer's guidelines. siRNA transfections were carried out using the transfection reagent Dharmafect1 (Dharmacon) following the manufacturer's instructions. For a full list of siRNA sequences see Supplementary Table 1.

## Plasmid generation

USP50 was amplified out of the addgene FLAG-USP50 plasmid vector (from Wade Harper's group[122]) and cloned into pcDNA5/FRT/TO. The pcDNA5/FRT/TO-FLAG-USP50 plasmids were designed and sent to Genscript for synthesis. These plasmids were made siRNA resistant to USP50 siRNA sequences 5 and 7 by introducing a series of silent point mutations as follows: USP50 siRNA sequence 5 - TAT GAT ACC CTT CCA GTT and corresponding siRNA resistant form - TAT GAC ACA CTA CCA GTT A and USP50 siRNA sequence 7−C TAC CCA GCA TTT ACG and corresponding siRNA resistant form - C TAT CCG GCT TTT ACG.

The pcDNA5/FRT/TO-Myc-FEN1 WT and E359K plasmids were synthesised by Genscript and include siResistance to FEN1 exon 2 siRNA GAUGCCUCUAUGAGCAUUUAU. Likewise, the pcDNA5/FRT/TO-EGFP-WRN WT, E84A and K577M plasmids were synthesised by Genscript and include siResistance to both WRN exon 9 siRNA GAGGGUUUCUAUCUUACUA and WRN exon17 siRNA AUACGUAACUCCAGAAUAC. The pCW-His-Myc-Ub plasmids were published previously[123].

## Inducible expression

Flp-InTM HeLa shUSP50-expressing cells were plated in 10 cm$^2$ dishes and transfected with a 4:1 ratio pcDNA5/FRT/TO-FLAG-USP50 variants and the pOG44 Flp Recombinase plasmid using FuGene6. Positive clones were selected with 100 μg/ml hygromycin and tested for expression of FLAG-USP50, by treatment with 2 μg/ml Dox for 72 hours and subsequent western blot.

Similarly, Flp-InTM HeLa shUSP50-expressing cells were transfected using FuGene6 with pcDNA5/FRT/TO-Myc-FEN1 variants or pcDNA5/FRT/TO-EGFP-WRN variants and pOG44 in a 4:1 ratio and positive clones selected with hygromycin 100 μg/ml. Expression of inducible genes was confirmed by western blot after incubation with 2 μg/ml Dox for 72 hours.

Plasmid and siRNA transfection FuGene6 was used to transfect DNA plasmids into cells at a 3:1 FuGENE (μl) ratio, following the manufacturer's guidelines. siRNA transfections were carried out using the transfection reagent Dharmafect1 following the manufacturer's guidelines. For details of siRNA sequences see Supplementary Table 1.

Antibodies details of antibodies used can be found in Supplementary Table 2.

## Colony survival assays

Colony survival assays were used to determine cellular sensitivity in response to HU or pyridostatin. Cells were plated at $2 \times 10^5$ cells/mL in a 24-well plate and treated with IPTG (100 μM) and/or Dox (2 μg/mL) to induce shUSP50 and FLAG-USP50 expression for 48 hours. Cells were treated with HU (16 hours) or pyridostatin (24 hours) before plating out in a six-well plate at low density. Plates were incubated for 14 days at 37 °C, 5% CO$_2$ until colonies formed. Colonies were stained using 0.5% Crystal violet in 50% methanol and colonies counted.

## Modelling

Molecular graphics and analyses were performed using UCSF ChimeraX, developed by the Resource for Biocomputing, Visualisation, and Informatics at the University of California, San Francisco, with support from National Institutes of Health R01-GM129325 and the Office of Cyber Infrastructure and Computational Biology, National Institute of Allergy and Infectious Diseases and Alphafold2 CoLab[124]. Sequences used were the first Ub from P0CG47 and USP50 sequence Q70EL3. Note this is not the reference sequence, NP_987090.2, which lacks the sequence "KFLLPS" found in isoform 2.

## Immunofluorescence and microscopy

Cells were plated in a 24-well plate on 13 mm circular glass coverslips at a density of $5 \times 10^4$ cells/ml. Cells were treated as required and then fixed in 4% paraformaldehyde (PFA) (unless otherwise stated). Once fixed, cells were permeabilized with 0.2% TritonX100 in 1× Phosphate-buffered saline (PBS), for 5 mins, blocked using 10% FBS in PBS for 5 mins and incubated with primary antibody for 1 hour at room temperature (RT) in 10% FBS/PBS (see Supplementary Table 2 for details). Cells were then washed in 10% FBS/PBS before being incubated for 1 hour with Alexa-Fluor antibodies at a concentration of 1:2000. Cells were washed in PBS and then fixed for 10 mins in 4% PFA before being washed again in PBS. DNA was stained using Hoechst at 1:20,000 for 5 mins and then washed with PBS before mounting onto Snowcoat slides using Immunomount mounting media. Cells were imaged on a Leica DM6000B microscope with an HBO lamp with a 100-W mercury short arc UV-bulb light source and four filter cubes, A4, L5, N3 and Y5, to produce excitations at wavelengths of 360, 488, 555, and 647 nm, respectively. Images were captured at each wavelength sequentially with a Plan apochromat HCX ×100/1.4 oil objective at a resolution of 1392 × 1040 pixels.

For EdU labelling of S-phase cells, cells were incubated with 10 μM EdU for 10 mins prior to fixation. EdU was then labelled with Alexafluor-647-azide using Click-IT technology. Briefly, permeabilised cells were incubated with the Click-IT reaction cocktail (PBS, 10 μM Biotin Azide, 10 mM Sodium Ascorbate, 1 mM CuSO4) for 30 mins at RT in the dark. Cells were washed with PBS containing 0.1% Tween 20 (PBST) and labelled with primary and secondary antibodies as above. For the assessment of symmetry, the ratio of the longest to shortest from a single origin was measured.

## Immunofluorescence and microscopy of ADP-ribosylation

Adherent cells grown on 13 mm circular glass coverslips (Thermo Fisher Scientific) were pre-extracted in 0.2% Triton x-100 in PBS for 2 mins on ice, to remove soluble nuclear content, and subsequently fixed with 4% formaldehyde in PBS for 10 mins at RT. When cells were stained for PCNA, an additional step was added after fixation: coverslips were treated with ice-cold methanol/acetone solution (1:1) for 5 mins at RT and washed 3 times for 5 mins in PBS. Thereafter, coverslips were blocked with 10% FCS in PBS for 1 hour at RT, followed by incubation with appropriate primary antibodies (1 hour at RT) and then incubation with fluorochrome-conjugated secondary antibodies (1 hour at RT). Coverslips were washed three times for 5 mins in PBST after both primary and secondary antibody incubations. Next, DNA was stained with DAPI (1 mg/ml in water) for 5 mins at RT and coverslips were mounted in fluoroshield (Sigma-Aldrich).

Automated multichannel widefield microscopy was performed using an Olympus ScanR Screening System equipped with an inverted motorised Olympus IX81 microscope and a motorised stage. Images were acquired using ×40 objective at a single autofocus-directed z-position under non-saturating settings. The inbuilt Olympus ScanR Image Analysis Software was used to analyse acquired images. Nuclei were identified by DAPI signal using an integrated intensity-based object detection module. The G1, S and G2 phase cells were gated based on PCNA and DAPI intensity, and fluorescence intensities of interest were quantified.

## Proximity ligation assay

Flp-InTM HeLa cells were seeded onto poly-L-lysine-coated coverslips. For EdU treatment, cells were pulsed with EdU for 10 mins (short-label) or 24 hours (long label) at 37 °C. For analysis of stalled forks, 5 mM HU was added into media following a short EdU pulse for 3 hours at 37 °C. Cells were pre-extracted for 5 mins on ice with Pre-extraction buffer (20 mM NaCl, 3 mM MgCl2, 300 mM Sucrose, 10 mM PIPES, 0.5% Triton X-100) and fixed in 4% PFA before blocking in 10% BSA overnight. The Click-IT reaction cocktail (PBS 1×, 10 μM Biotin Azide, 10 mM Sodium Ascorbate, 1 mM CuSO$_4$) was added for 1 hour at RT in the dark. The Click-IT reaction cocktail was then removed, and cells were incubated in blocking solution for a further 30 mins before incubation with primary antibodies (details in Supplementary Table 2) in 10% FBS in PBS for 1 hour at RT in the dark. After incubation with primary antibodies, cells were incubated with the corresponding MINUS/PLUS PLA probes (Sigma DUOlink PLA kit) for 1 hour at 37 °C in a warm foil-covered box. This was after 2 washes in Buffer A (Sigma DUOlink PLA kit) and then incubated with the PLA kit Ligation solution (1× ligation buffer, ligase enzyme) for 30 mins at 37 °C. Cells were again washed again in wash buffer A before incubation for 100 mins at 37 °C with the PLA kit amplification solution (1× amplification buffer, polymerase enzyme). Following amplification cells were washed for 15 mins with wash buffer B (Sigma DUOlink PLA kit) and incubated with Hoechst for 5 mins before another 15 mins wash with buffer B. A final 60 secs wash in 0.01% wash buffer B was performed. Coverslips were mounted onto glass slides and imaged and quantified the following day using a Leica DM6000 fluorescent microscope with a ×100 objective lens.

## FLAG immunoprecipitation

Flp-InTM HeLa FLAG-USP50 cells were plated in a 10 cm plate and treated with Dox for 72 hours to express inducible FLAG-USP50. Cells were washed with 10 ml ice-cold 1x PBS before being scraped in ice-cold Nuclear Lysis Buffer (10 mM HEPES pH 7.6, 200 mM NaCl, 1.5 mM MgCl2, 10% Glycerol, 0.2 mM EDTA, 1% Triton) for every 10 ml, 1 protease inhibitor tablet (cOmplete – SIGMA), 1 phosphatase tablet (PhosSTOP – Roche), 20 μM MG132, 1 μl DNase1 and 200 μl iodoacetamide was added. The lysed cells were then transferred into 1.5 ml Eppendorf tubes and incubated with the nuclear lysis buffer on ice for 1 hour with rotation before centrifugation at 16,000 × $g$, 4 °C for 10 mins and the supernatant was kept, and the pellet was discarded. 50 μl of the supernatant was mixed with 20 μl 4× SDS Loading buffer and boiled at 95 °C for 5 mins. For every IP, 10 μl FLAG-agarose beads were firstly washed out of storage buffer by doing 3× 1 ml PBS washes and centrifuging at 800 × $g$ rpm between each wash. 60 μl of binding buffer (PBS and nuclear lysis buffer at a ratio of 2:1.5) was added for every 10 μl of agarose beads. The re-suspended beads were then added to 450 μl of supernatant for each sample and rotated at 4 °C overnight. The following day, the samples were centrifuged at 800 × $g$ for 60 secs and the beads left to settle. The supernatant was removed before 3 × 1 ml PBST washes. The wash buffer was completely removed before adding 60 μl 2× SDS loading buffer. This was boiled at 95 °C for 5 mins and 10 μl loaded onto an SDS PAGE gel and analysed by Western blotting.

## Fibre labelling and spreading

Cells were seeded in 6 cm plates and treated for 72 hours to knock down or overexpress proteins of interest and then treated with thymidine analogues. To monitor ongoing replication dynamics, cells were incubated at 37 °C with 5-chloro-2′-deoxyuridine (CldU) for 20 mins at a final concentration of 25 μM and then with CO$_2$-equilibrated 5-Iodo-2′-deoxyuridine (IdU) at 37 °C for 20 mins at a final concentration of 250 μM. After incubation with the thymidine analogues, cells were washed once with ice-cold PBS, trypsinized and re-suspended in 200 μl of PBS and counted. The optimal final cell density is 50 × 104 cells/ml and thus cells were adjusted to reach such a concentration. For each sample, three Snowcoat slides were labelled. Near the label of each slide 2 μl of the cell, sample was placed and allowed to slightly dry for 3–4 mins. Then 7 μl of spreading buffer (200 mM Tris pH 7.4, 50 mM EDTA, 0.5% SDS) was added, mixed with the sample, and incubated for 2 mins. To spread the sample down the slide, slides were gradually tilted and once the sample had reached the bottom of the slide, they were allowed to dry for 2 mins. Finally, slides were fixed in a 3:1 ratio of Methanol: Acetic acid for 10 mins before leaving slides to air dry for 5–10 mins. Dried slides were stored at 4 °C till staining.

## Fibre immunostaining

After fibre spreading slides were washed 2× for 5 mins with 1 ml H2O and rinsed with 2.5 M HCl before denaturing DNA with 2.5 M HCl for 1 hour 15 mins. Slides were then rinsed 2× with PBS and washed for 5 mins in blocking solution (PBS, 1% BSA, 0.1% Tween 20). Slides were incubated for 1 hour in blocking solution. After blocking, each slide was incubated with 115 μl of primary antibodies, Rat αBrdU (Abcam) used at a concentration of 1:1000 and Mouse αBrdU (BD Biosciences) used at 1:750. Slides were covered with large coverslips and incubated with the antibodies for 1 hour. After incubation with the primary antibody, slides were rinsed 3× with PBS and then incubated for 60 secs, 5 mins and 25 mins, with blocking solution. After rinsing and washing, slides were incubated with 115 μl of secondary antibodies (α-Rat AlexaFluor 555 and α-Mouse AlexaFluor 488) in blocking solution, at a concentration of 1:500, covered with a large coverslip for 2 hours. Slides were rinsed 3× with PBS and incubated with blocking solution for 60 secs, 5 mins and 25 mins. After again rinsing 2× with PBS mounting media was added to the slide and a large coverslip placed over the slide and it was left to dry. Coverslips were then stored at −20 °C for microscopy analysis. It is important to point out that during this process slides were kept protected from light.

## Fibre scoring and analysis

Using 10 image fields of fibres per condition, all visible fibres were assigned a structure (ongoing fork, 1st label origin, 2nd label origin, 1st label termination, 2nd label termination). Then the proportion of each structure was determined based on the total amount of all scored structures containing a first label. For fork asymmetry, only first-label origin forks were studied. Both IdU-second-label lengths were measured from each side of the CldU first label using the Fiji/ImageJ software version 1.4.3.67[125,126], and the lengths were compared and given as a proportion of the longest label.

## Single-molecule analysis of resection tracks (SMART)

were performed as previously described[127]. Briefly, cells were seeded in 6 cm plates and treated for 72 hours to knock down or overexpress proteins of interest. Cells were additionally pulsed with 20 μM BrdU for these 72 hours and treated with 5 mM HU for the last 3 hours. After incubation with HU, cells were harvested, and fibre spreads were prepared as described above. Native fibre spreads were subsequently stained as described above, omitting HCl treatment and extending the initial blocking step to 2 h. Native BrdU tracks were immunostained using Mouse αBrdU (BD Biosciences) and α-Mouse AlexaFluor 488 antibodies both at 1:500 dilutions. To quantify ssDNA resection track size, the lengths of green (AF 488) labelled native patches were measured using ImageJ and subsequently converted to μm.

## INDUCE-seq DSB mapping and sequence analysis

Cells were harvested in Dulbecco's PBS and counted. Cells (120,000 per well) were adhered onto a poly-L-lysine-coated 96-well plate and cross-linked in methanol-free paraformaldehyde (final concentration 4%) for 10 mins. The PFA was removed, and the wells were washed twice in PBS and stored in 200 μl PBS. Plates were sealed and stored at 4 °C until downstream library preparation. INDUCE-seq was

performed as previously described[58] on Illumina NextSeq500 using 1× 75 bp high-capacity flow cell. INDUCE-seq was performed in duplicate. After assessing reproducibility by comparing the genome-wide densities of DSBs in 10-kb windows, technical replicates were combined. INDUCE-seq reads were processed as previously described and aligned to the human genome with bowtie2 (GRCh38/hg38)[128]. Using Bedtools[129], alignments were converted to Bed files and inter-sects between biological repeats generated. These were used to generate fasta sequences using Getfasta. Duplicate sequences were removed by Filter Fasta. Nucleotide% were displayed using Fasta Statistics. Oligo-diff was then run comparing the USP50 and siNTC data sets to return oligos significantly enriched in one file relative to the other[130].

## CO-FISH
The protocol was based on refs. 131. In brief, HeLa cells were grown in six-well tissue culture dishes in the presence or absence of 1 mM IPTG, to induce shUSP50 expression and Doxycyclin to induce wild-type or mutant USP50 expression. HeLa cells carrying the expression cassette for WT or mutant USP50 were grown with Tetracyclin-free FBS (pan biotec, p30 3602). Transfection with siRNA for the respective Helicases was performed with Dharmafect1 (Dharmacon) according to the manufacturer's instructions. On the following day, the Telomerase inhibitor Bibr1532 was added to a final concentration of 20 μM (Cambridge Biosciences CAY16608). After 4 days, BrdU was added to a final concentration of 2.5 μM, and cells were incubated for 16 h before the addition of Colcemid (Gibco 15212012) to a final concentration of 0.2 μg/ml. After 4 h cells were trypsinized and re-suspended in 75 mM KCl and incubated for 30 mins in a water bath at 37 °C. Cells were pre-fixed by the addition of 1 ml of 3:1 methanol/acetic acid, pelleted and fixed with 4 ml of 3:1 methanol/acetic acid and stored at −20 °C. Prior to dropping slides, cells were pelleted and re-suspended in fresh 3:1 methanol/acetic acid, then dropped on microscopy slides to produce metaphase spreads. The following day, slides were incubated with RNase A (100 μg/mL) at 37 °C for 10 min, rinsed with PBS and post-fixed with 4% formaldehyde at RT for 5 min, rinsed 3× with PBS, incubated with 0.1 M HCl for 20 mins at RT, rinsed with water, dehydrated in an ethanol series (15%, 85%, 100% cold ethanol 2 mins each) and air-dried. Slides were then incubated with 0.5 μg/ml Hoechst 33258 in 2× saline sodium citrate (SSC) (Sigma) for 15 mins and exposed to UV (UVP Crosslinker CL-3000, Analytik Jena) for 30 mins in 2× SSC before being treated with exonuclease III (3 U/μl) for 10 mins. Slides were rinsed with water, dehydrated in an ethanol series (15%, 85%, 100% cold ethanol 2 mins each) and air-dried. Metaphases were incubated with pre-hybridisation buffer (70% formamide, 0.1% Tween 20) for 1 h at RT. The pre-hybridisation buffer was replaced with hybridisation cocktail (0.4 μg/mL fluorochrome-labelled PNA Alexa 488–OO-CCCTAA (Eurogentec PN-TC060-005)), 70% formamide, 0.1% Tween 20, 10% dextran sulfate), metaphases were denatured at 78 °C in 2× SSC solution for 10 mins before hybridisation at 37 °C for 16 hours. Metaphases were washed 2 × 10 mins in 2× SSC at 42 °C; the slides were then rinsed in PBS and mounted with Fluoromount before imaging on a Zeiss Axio Observer widefield system with a ×100 1.4 NA oil objective, Hamamatsu ORCA Flash 4.0 camera and LED Colibri.2 illumination. Z-stacks consisting of 11 slices separated by 350 nm were acquired to ensure capture of all telomere signals. Telomeres were scored with the Multipoint selection tool in Fiji/ImageJ software version 1.4.3.67[125].

## Chromatin fractionation
To separate the chromatin-enriched fraction, cells were harvested and washed in PBS, before being re-suspended in sucrose buffer (10 mM Tris-Cl pH 7.5, 20 mM KCl, 250 mM sucrose, 2.5 mM MgCl$_2$, protease inhibitor). Triton X-100 was added to a final concentration of 0.3%, and cells were vortexed 3 × 5 s, followed by centrifugation (500 × $g$, 4 °C, 5 mins). The supernatant was discarded, and the pellet re-suspended in

NETN150 buffer (50 mM Tris-Cl pH 8.0, 150 m NaCl, 2 mM EDTA, 0.5% NP-40, protease inhibitor) and incubated on ice for 30 mins, followed by centrifugation (1700 × $g$, 4 °C, 5 mins). The supernatant was discarded, and the pellet re-suspended in NETN150 buffer and sonicated 2 × 10 s. Subsequently, 2× SDS loading buffer was added and the samples were boiled before analysis by western blotting.

## iPOND
iPOND was performed as described previously[70]. Briefly, cells were treated with 10 μM EdU for 10 mins, with or without a 5 mM HU treatment for 3 h. Cells were then cross-linked with 1% paraformaldehyde for 20 mins, washed with PBS and permeabilised with 0.25% Triton X-100 in PBS for 30 mins. Following further PBS washes, cells were incubated in click reaction buffer (10 mM sodium ascorbate, 2 mM CuSO4, 10 μM biotin azide in PBS) for 2 h. Following further PBS washes, cells were lysed in 1% SDS buffer, and extensively sonicated. Following a clarifying centrifugation (13,000 × $g$, 4 °C, 10 mins), and dilution by half in PBS, lysates were incubated with biotin-agarose beads overnight. Beads were washed once in lysis buffer, once in 1 M NaCl, then twice more in lysis buffer, before elution in 4× SDS loading buffer with boiling.

## In vitro pulldown assay
Bacterial cultures were grown at 37 °C for 6 hours before overnight induction at 25 °C by treating with 1 mM IPTG. Cultures were spun down, and pellets were lysed in ice-cold NP20 lysis buffer (50 mM Tris pH 7.4, 150 mM NaCl, 1% NP20) supplemented with EDTA-free protease inhibitors (1 tablet in 10 mls). The lysates were then sonicated (3 × 30 secs with 60 secs between each prep on power 5) before 10 μl benzonase was added and rotated at 4 °C for 30 mins. Lysates were then centrifuged at 20,000 × $g$ for 40 mins. The supernatant was kept on ice, and 500 μl of 50% beads per sample were added. This was incubated at 4 °C overnight then kept on ice during 3 × 15 mins washes in lysis buffer with agitation before being re-suspended in 100 μl of 2× SDS loading buffer. 10 μl of the sample was then run on a 10% SDS gel.

## Reporting summary
Further information on research design is available in the Nature Portfolio Reporting Summary linked to this article.

## Data availability
The sequencing data generated in this study have been deposited in NCBI's Gene Expression Omnibus under GEO Series accession code GSE269605. Source data are available in Figshare at https://doi.org/10.6084/m9.figshare.26005390.

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

## Acknowledgements

Grant funding was as follows. CRUK: C8820/A19062 and A28283 (H.L.M., H.R.S., G.R., K.S., and J.B.), Breast Cancer Now: 2015MayPR499 (K.S.) and Studentship 2010NovPhD02 (H.R.S.). Wellcome Trust 206343/Z/17/Z (A.W. and A.G.), CRUK Centre training (K.E.), and University of Birmingham (S.V.). Worldwide Cancer Research 13-1048 (E.P.) and Medical Research Council MR/J007595/1 (E.P.). FLAG-HA-USP50 was a gift from Wade Harper (Addgene plasmid # 22588; http://n2t.net/addgene:22588; RRID:Addgene22588)[122]. Nick Davies provided expert advice for the CO-FISH experiments. We thank Professor Andrew Beggs, University of Birmingham for HCT116 and RKO cells, and Professor Keith Caldecott for discussions and reading the manuscript. We thank the Microscopy and Imaging Services at Birmingham University (MISBU) and the UoB Flow Cytometry Services (UoBFC) in the Tech Hub facility for microscope and FACS support and maintenance.

## Author contributions

H.R.S. generated reagents, fiber experiments, and in vitro analysis.
H.L.M. performed fibre assays, chromatin fractionations, foci analysis,

immunoprecipitations, and P.L.A. analysis. G.E.R. performed chromatin fractionation assays. K.E. performed fiber experiments, chromatin fractionations, and colony assays. A.L. performed colony assays and PLA analysis. Y.A. conducted fork restart assays and PLA analysis. K.S. and A.K.W. performed PLA analysis. A.J.G. generated reagents, performed cell survival and foci analysis. F.D. and P.V.E. performed INDUCE-Seq. S.A.K. and R.M.D. performed CO-FISH analysis. E.J.A. conducted fork restart assays. A.L.P. performed native BrDU assays. A.S.C. conduction immunoprecipitations. P.C.T. performed foci analysis. A.V. performed the PARGi experiments. S.V. generated reagents and performed foci experiments. J.B. generated proteins and performed binding assays. E.P. advised on fiber analysis. E.J.B. advised H.R.S., S.H.R. supervised F.D. and P.V.E., M.S. advised on bioinformatics. J.R.M., H.M., and R.M.D. wrote the paper. All authors commented on the paper and research. J.R.M. conceived and directed the project.

## Competing interests

The authors declare no competing interests.

## Additional information

[1]Birmingham Centre for Genome Biology and Institute of Cancer and Genomic Sciences, University of Birmingham, Birmingham B15 2TT, UK. [2]Broken String Biosciences Ltd., BioData Innovation Centre, Unit AB3-03, Level 3, Wellcome Genome Campus, Hinxton, Cambridge CB10 1DR, UK. [3]Division of Cancer & Genetics School of Medicine, Cardiff University, Heath Park, Cardiff CF14 4XN, UK. [4]Genome Damage and Stability Centre, School of Life Sciences, University of Sussex, Falmer, Brighton BN1 9RQ, UK. [5]Abramson Family Cancer Research Institute, Perelman School of Medicine, University of Pennsylvania, Philadelphia421 Curie Boulevard PA, 19104-6160, USA. [6]Present address: CCTT-C Cancer Research UK, Clinical trials unit, Sir Robert Aitken building, College of Medicine and Health, University of Birmingham, Birmingham B15 2TT, UK. [7]Present address: School of Chemical, Materials and Biological Engineering, University of Sheffield, Mappin Street, Sheffield S1 3JD, UK. [8]Present address: Adthera Bio, Lyndon House, 62 Hagley Road, Birmingham B16 8PE, UK. [9]Present address: SUMO Biology Lab, School of Molecular and Cellular Biology, Faculty of Biological Sciences, University of Leeds, Leeds LS2 9JT, UK. [10]Present address: Department of Genetics and Development, Herbert Irving Comprehensive Cancer Center, Columbia University Irving Medical Center, New York, NY, USA. [11]Present address: University Hospital Birmingham N.H.S. Foundation Trust, Queen Elizabeth Hospital Birmingham, Mindelsohn Way, Birmingham B15 2TH, UK. [12]These authors contributed equally: Hannah L. Mackay, Helen R. Stone. ✉e-mail: j.morris.3@bham.ac.uk

