## [Peer Review File · Nature Communications]

USP50 suppresses alternative RecQ helicase use and deleterious DNA2 activity during replication.REVIEWER COMMENTS

Reviewer #1 (Remarks to the Author):

The manuscript describes a study showing that a Ub-mediated pathway that USP50 influences WRN, FEN1, DNA2, RECQL4/5 at ongoing and stalled replication forks. However, these findings don't represent a sufficiently advance and it currently lacks depth of mechanistic advance. I don't think this manuscript is suitable for NC.

1, Recent work on USP50 suggests that USP50 reduces Ku70 or ACE2 protein levels by promoting Ku70 or ACE2 degradation. Moreover, deubiquitinase-inactive mutant of USP50 (USP50-C53S) lost the ability to increase ACE2 protein levels, suggesting that USP50 regulates ACE2 levels in a manner dependent on its deubiquitinase activity [1,2]. However, author showed that USP50 is an inactive ubiquitin-specific protease and it lacks the conserved acidic residue of the catalytic triad. Is USP50 really an inactive ubiquitin-specific protease?

Ref:

1, Zuo Y, Zheng Z, Huang Y, He J, Zang L, Ren T, Cao X, Miao Y, Yuan Y, Liu Y, Ma F, Dai J, Tian S, Ding Q, Zheng H. Vitamin C promotes ACE2 degradation and protects against SARS-CoV-2 infection. *EMBO Rep.* 2023 Apr 5;24(4):e56374.

2, Cai J, Wei J, Schrott V, Zhao J, Bullock G, Zhao Y. Induction of deubiquitinating enzyme USP50 during erythropoiesis and its potential role in the regulation of Ku70 stability. *J Investig Med.* 2018 Jan;66(1):1-6.

2. In Fig1, The author hypothesized that USP50 bind to UB and enriched at stalled replication fork. In Fig1B, a high MW (far away above 170KD) myc-ub blotting band is shown and the author clarified it is a high molecular weight Ub conjugates which bind to USP50. what is this Ub conjugates? How to exclude that it is the USP50 itself ubiquitination modification? If USP50 bind to UB is ture, another important question is it only happen at the stalled fork or all through the chromatin? In Fig1D, the same question, we can argue that MG132 blocked the proteasome dependent USP25 degradation and more USP25 ubiquitination modification was shown. iPOND assay was needed to confirm USP25 WT but not IR mutant as recruited at the stalled fork.

3. In fig 2E-G, fork stalled mediated DSBs typically repaired by HR and USP50 is reported to stabilize KU70, 53BP1 is not a good maker here, g-H2AX and BRCA1 foci should be examined here.

4. For Fig 3, does USP50 deubiquitinate WRN? What is the mechanism that USP50 promote WRN to load to replication fork (should be confirmed by iPOND), bind to Fen1 and 9-1-1?

5. Another big gas here is that why and how WRN/FEN1 switch to RECQL/DNA2 at the fork when USP50 is depleted? What is the role of USP50 in this process?

6. Most of the knockdown strategies utilizing in the manuscript are only one target, either siRNA or shRNA, another si/shRNA should be used to exclude the off target effect.

7. In MSI-H colorectal cell lines, WRN loss results in severe genome integrity defects and decreased viability. However, in Fig 3H HCT116 cells, depleted of USP50 doesn't affect cell viability but WRN depletion show dramatic effect? Is that suggest USP50 doesn't regulates WRN function here?

Reviewer #2 (Remarks to the Author):

In this manuscript, Morris and co-workers describe a protective role for USP50, a poorly understood and catalytically-inactive deubiquitinase (DUB), at DNA replication forks (RFs) required for suppressing DSB formation at specific genomic loci. As such the work is original and of considerable interest to the DNA repair and ubiquitin signaling community. Overall, most of their data is consistent with the view that USP50 localizes to chromatin in a ubiquitin(Ub)-binding-dependent manner (substrate(s) unknown) to promote the preferential use of DNA helicase:nuclease complexes (WRN:FEN1 over RECQL4/5:DNA2) at unperturbed and stressed RFs. However, there are quite some inconsistent (or missing) data and technical issues in the manuscript requiring major revisions that must be addressed before I can recommend it for publication.

1) Why did the authors choose to work on USP50 in the first place? It is stated (lane 99) that “Two previous RNA interference screens have highlighted USP50 as potentially important to replication.” In the first study, the Morris lab perform an siRNA screen to identify DUBs implicated in the clearance of Ub conjugates after release from hydroxyurea (HU) (Butler et al., 2012). In this paper, however, there is no screen data included. Therefore, I would ask the authors to provide this data. The second study provides at least the siRNA screen data showing that USP50 depletion does results in reduced cellular survival upon HU treatment.

2) In Figure 1, there is quite some additional experimental data required to support USP50 binding to Ub conjugates via a conserved pocket (I141) is promoting its localization to stalled RFs:

- o To show that USP50 binding to Ub conjugates supports USP50 chromatin association, they should include Ub-I44A mutant (Figure 1B) and an E1 inhibitor (Figures 1B and 1D).
- o Figure 1C is redundant with and less convincing than Figure 1F, and could be easily removed.
- o Data shown in Figure 1D raises many questions: Is USP50 itself conjugated with ubiquitin, (i.e., polyubiquitinated) and perhaps degraded on chromatin? Why did VCPi not work in the same manner as MG132 (I suggest prolonging treatment with VCPi to 6 hours according to Anderson, Cancer Cell 2015)? Finally, also this experiment should be conducted with the I141R mutant and with an E1 inhibitor to substantiate the conclusions.
- o The experimental set up of Figure 1E (EdU treatment was for 24 hours) is quite unusual. In principle a co-localization of USP50 with PCNA foci would be more informative. There is no concluding statement in the text for this data. I would suggest removing but include in Figure 1G (EdU pulse) samples without HU treatment to unambiguously illustrate “USP50 enrichment at stalled forks”.
- o Figure 1F highlights an important finding, yet the western blots are extremely small and the signal for USP50 quite weak.

3) In Figure 2, the authors design a stable cell system where the shRNA against endogenous USP50 as well as the expression of FLAG-tagged USP50 can be induced. Even though it may be true that HeLa cells express very low levels of USP50, I would still highly recommend confirming USP50 depletion at least at the mRNA level using RT-PCR. In fact, it seems as if expression of the USP50-I141R mutant has a dominant-negative effect (in absence of shUSP50 expression), suggesting that the phenotypes seen with shUSP50 expression could be caused by an off-target effect. One might

wonder whether depletion of a factor that is expressed at undetectable levels can produce any specific phenotype? Therefore, it is crucial to confirm at least a few results with a different RNAi-targeting sequence (i.e., with a second shRNA against USP50)

4) In Figure 2D, and again in Figure 4F, attempting to provide experimental evidence for increased ssDNA formation in response to HU treatment in USP50-depleted cells, the authors show very poor and unconvincing western blot data for phosphorylated RPA2 (as a ssDNA marker). These blots have either to be significantly improved (total RPA2 is also missing in Figure 4F) or an alternative readout should be used. For instance, others have employed immunofluorescence microscopy analysis of BrdU staining under non-denaturing conditions after HU treatment (e.g., Leung et al Cell Reports 2023).

5) Increased 53BP1 foci formation and reduced survival following replication stress (induced by HU and PDS) could also indicate a defect in HR-mediated fork restart in cells depleted of USP50. It could be informative to examine whether USP50 depletion impairs HR using classical DR-GFP reporter cells.

6) Data in Figure 3D suggesting USP50-dependent interaction between WRN and FEN1 (Figure 3D) based on in situ proximity assays are not convincing (no representative images are shown) and must be corroborated with co-IP data.

7) Figures 3E-G highlight the functional interplay between USP50 and WRN:FEN1 at both unperturbed and HU-challenged RFs. However, they are all based on different experimental readouts (53BP1 foci after HU, ongoing replication using different DNA fiber labelling schemes). To corroborate these important mechanistic findings, I would strongly recommend using for all conditions (overexpression of WRN and FEN1 variants) the same readouts without HU (e.g., 1st label termination or fork asymmetry) and with HU (% stalled forks (3A) and 53BP1 foci (3E)).

8) Figure 3I indicates a role for USP50 in telomere maintenance. Is this function dependent on Ub-binding (use IR mutant) and on USP50-mediated WRN-FEN1 activity at RFs (use siWRN and siFEN1 (as in 3A) and/or WRN WT and mutant, FEN1 WT and mutant expression constructs (shown in Figure 3F and 3G))?

9) The increase in DNA2 foci numbers following USP50 depletion in HU-treated cells is very minimal and barely visible (Figure 4A, average foci number per cell increases from 12 to 15) and could simply be due to a slight increase in DNA2 protein levels (Suppl Figure 3I). I suggest checking DNA2 levels on chromatin after HU in control- and USP50-depleted cells by western blotting to substantiate the microscopy data.

Similarly, could there be an increase of RECL4/5 levels at chromatin after HU upon USP50 depletion (like what was observed for WRN (Figure 3B))?

10) It would be quite revealing to test whether USP50 'directly' associates with any of the nuclease helicase factors in an UB-dependent manner? Along these lines, perhaps USP50 regulates the resection of certain specific DSBs via controlling the access of WRN, thereby rejecting DNA2, two

proteins implicated in DSB resection.

Minor comments:

- I assume that in Figure 2E, HU treatment is mis-labeled as EdU+.
- Include HU treatment in the Figure legends for panels 2F, 2G and 3E.

Reviewer #3 (Remarks to the Author):

In this manuscript, the authors employed human cell lines and various cell biological and biochemical techniques to investigate the role of USP50 in DNA replication. They demonstrated that the inactivation of USP50 affects cell survival in response to HU-induced replication stress, yet the mechanistic insight into its role in DNA replication remains elusive. Based on their experiments, predominantly utilizing the DNA fiber assay, the authors propose a model in which USP50 binds to ubiquitinated substrate X on chromatin at the sites of DNA replication forks/stalled forks, thereby stimulating the recruitment of the WRN and FEN1 complex, which coordinates unperturbed DNA synthesis and fork restart. In contrast, the absence of USP50 results in reduced WRN:FEN1 complex presence at stalled replication fork/chromatin sites, allowing for an increased presence of DNA2, RecQL4/5, whose nuclease and helicase activities negatively affect the stability and restart of DNA replication forks, particularly at some GC-rich regions.

This is a well-structured manuscript; however, it requires additional confirmation of a few key conclusions for publication in a reputable scientific journal. Two critical points need addressing:

First, the authors must provide direct evidence of USP50 inactivation leading to a reduction in the levels of WRN and FEN1 proteins at the replication forks, simultaneously with an increase in the levels of DNA2 and RecQL4/5. While Figure 4A shows increased DNA2 foci in USP50 cells, this effect should be substantiated through iPOND technology or by examining isolated chromatin from S-phase cells to elegantly demonstrate these changes.

Secondly, the authors should validate their model by assessing its impact on cell survival. The diminished survival of USP50-deficient cells in response to HU and Pyr, as observed in Figure 2H and I, respectively, should be rescued when DNA2 or RecQL4/5 is co-inactivated or co-depleted.

Specific comments:

1. Figure 1B: The authors should also analyze the total ubiquitylation signal on the membranes.

2. Figure 1D: The 5 μ M/4hr MG132 treatment should deplete the nuclear pool of ubiquitin. If their model is correct, USP50 recruitment to chromatin should not occur in the absence of ubiquitin in the nucleus. The results in this figure suggest that USP50 chromatin removal depends on the proteasome.

3. Figure 2D: The authors should also demonstrate the restoration of ssDNA formation through the overexpression of USP50-wt but not USP50-IR.

4. Figure 3B: This is a crucial experiment for their model. However, the WRN signal on the Western blot is weak. The authors need to improve the quality of this experiment and determine whether this phenotype is restored with the overexpression of USP50wt and USP50-IR. As suggested earlier in this report, the iPOND assay is the best method to demonstrate that WRN is reduced at stalled DNA replication forks in USP50-depleted cells.

5. Figure 4A and K: The quality of the foci images should be improved.

6. As previously suggested, the authors need to demonstrate an increased amount of DNA2 and RecQL4/5 protein at the fork (using iPOND) or replicative chromatin (in S-phase) in USP50-depleted cells. They should then show the restoration of this phenotype with the overexpression of USP50 wt, but not USP50-IR.

7. The authors should use a CFA assay as in Figure 2H or I to measure cell survival in USP50-depleted cells and demonstrate rescue through co-depletion of DNA2 or RecQL4/5 and expression of WRN-wt, but not WRN-enzymatically inactive variants.

Minor points and suggestions:

1. Instead of using a CldU-HU-IdU assay to measure stalled forks, consider using this assay to measure fork restart.

2. It is not clear how reference 41 (Butler et al., 2012) highlights the role of USP50 in DNA replication. The authors should clarify whether siUSP50 increases or decreases the total Ub/FK2-signal based on the original paper.

3. Investigate well-known E3-ubiquitin ligases involved in DNA replication and replication stress response, such as RNF8, RNF168, BRCA1/BARD1, and Cullin-RING ligases, to support the Ub-based model. For cullin-RING ligases, utilizing a NEDD8 inhibitor may be sufficient.

4. Supplementary Figure 4 should be Supplementary Figure 1.

Response to reviewers' comments NCOMMS-23-37261

Reviewer #1:

The manuscript describes a study showing that a Ub-mediated pathway that USP50 influences WRN, FEN1, DNA2, RECQL4/5 at ongoing and stalled replication forks. However, these findings don't represent a sufficiently advance and it currently lacks depth of mechanistic advance. I don't think this manuscript is suitable for NC.

An overview of our findings are as follows:

Evidence that USP50, independent of any canonical DUB behaviour (see below for further detail) regulates replication dynamics.

USP50 does so through promoting FEN1 and WRN presence at stalled replication forks.

In the absence of USP50, DNA2, RECQL4/5 act to suppress normal replication kinetics. These proteins are responsible for HU- and prydostatin sensitivity and the loss of telomeres in USP50-depleted cells. These proteins are also responsible for the poor fork restart of FEN1-depleted cells.

Advance: illustration that nuclease/helicase use is actively restricted, that inappropriate nuclease/helicase engagement is deleterious and that the restriction to the correct enzymes requires the inactive DUB, USP50.

These findings fulfil the aims and scope of the journal, which are to publish "*high-quality research... represent[ing] important advances of significance to specialists within each field*".

1, Recent work on USP50 suggests that USP50 reduces Ku70 or ACE2 protein levels by promoting Ku70 or ACE2 degradation. Moreover, deubiquitinase-inactive mutant of USP50 (USP50-C53S) lost the ability to increase ACE2 protein levels, suggesting that USP50 regulates ACE2 levels in a manner dependent on its deubiquitinase activity [1,2]. However, author showed that USP50 is an inactive ubiquitin-specific protease and it lacks the conserved acidic residue of the catalytic triad. Is USP50 really an inactive ubiquitin-specific protease?

We have performed several further experiments to test this question. We generated a new mutant of USP50, substituting the remaining two residues of the USP catalytic triad. i.e. in addition to lacking the Asp/Asn residues, we also substituted the remaining two possible catalytic residues, C53S and H327A. We compared the expression of this mutant, finding that it behaved similarly to WT-USP50, suppressing sensitivity to hydroxyurea and suppressing the generation of spontaneous 53BP1 foci (Supplemental Figure 3 I-K and O). We state in the discussion that while we cannot comment on USP50's role in regulating Ku70 or ACE2, our data clearly indicate that no canonical USP-activity is important to USP50's role in suppressing markers of DNA breaks or supporting cell survival in the presence of hydroxyurea.

2. In Fig1, The author hypothesized that USP50 bind to UB and enriched at stalled replication fork. In Fig1B, a high MW (far away above 170KD) myc-ub blotting band is shown and the author clarified it is a high molecular weight Ub conjugates which bind to USP50. what is this Ub conjugates? How to exclude that it is the USP50 itself ubiquitination modification? If USP50 bind to UB is ture, another important question is it only happen at the stalled fork or all through the chromatin? In Fig1D, the same question, we can argue that MG132 blocked the proteasome dependent USP25 degradation and more USP25 ubiquitination modification was shown. iPOND assay was needed to confirm USP25 WT but not IR mutant as recruited at the stalled fork.

To address whether USP50 can bind Ub we have performed several new experiments. We generated USP50 in bacteria and found that it could co-purify Ub-conjugates (Supplemental Figure 2B). This excludes ubiquitination of USP50 itself in the binding of Ub. We do appreciate that some of the bands in 1B (now Figure 1C) are ubiquitinated USP50, nevertheless the use of the IR-mutant indicates a proportion of the pull-down is dependent on the ile-141 face, given that this mutant is of a hydrophobic residue of the pocket and not of a lysine, where ubiquitination occurs. We agree with the reviewer that USP50 is itself Ub-modified, as suggested by the high molecular weight bands found after blocking the proteasome with MG132. To exclude USP50 degradation effects, we quantified the impact of MG132 on USP50 protein levels, compared to USP50 levels at chromatin, finding the impact on chromatin association far greater than the impact on USP50 protein levels. We have highlighted the impact of MG132 in the text.

We have performed IPOND to do as the reviewer suggested— and have been successful with FEN1 and PCNA (Figure 3I). However, we have had no success with larger proteins (WRN, RECQL4, RECLQ5, USP50), despite the known presence of some of them at replication forks. We attempted to tackle the known drawback of poor HMW-protein isolation in IPOND by using several variants of the protocol that claim to overcome the problem (RIPA buffer and diluted SDS¹ and native²), plus several variants of our own, but without success (blots at the base of this response show a series of attempts). Thus, the question of whether it is chromatin and/or the stalled replication fork that recruits USP50 was tested using alternative types of measurement: for chromatin assessment: fractionation of chromatin and proximity-linked ligation with EdU after 24-hour incubation, to label all DNA (ie. chromatin), and labelling only nascent DNA at stalled forks using short EdU incubation (Figure 1F). The proximity-linked ligation assays use specific antibodies to address whether the labelled DNA occurs within 40 nm of tagged USP50 (the limit for PLA)³. The PLA used in the way we have employed it and iPOND are not entirely equivalent, they are certainly highly related, both using interaction with incorporated nucleotide analogue as a measure of protein engagement. Indeed the same PLA approach, using short-term EdU incorporation, has been used in recent investigations by others, for example, to investigate MRE11 and EXO1 interaction with nascent DNA at the stalled replication forks, Nusawardhana, *et al.* Nucleic acid research 2024⁴ and to assess the localization of the newly identified reversal factor TFIPII with nascent DNA at forks, Chen *et al* Nature Comms 2024,⁵. To test it further we used very short EdU incorporation times (5 mins) and also performed a thymidine chase with excess thymidine for 10 minutes before HU treatment (Figure 1F). This experiment confirmed the requirement on nascent DNA to induce the signal. We have been careful not to overclaim and state our data shows USP50 “at or near” nascent DNA.

3. In fig 2E-G, fork stalled mediated DSBs typically repaired by HR and USP50 is reported to stabilize KU70, 53BP1 is not a good maker here, g-H2AX and BRCA1 foci should be examined here.

As suggested we examined BRCA1, see Supplemental Figure 3E and Figure R1.

Figure R1. BRCA1 foci after shUSP50 / siMUS81 treatment. BRCA1 foci numbers in HeLa cells treated with siNTC or shUSP50 and co-treatment or not with siRNA targeting MUS81. Data is from 3 independent experiments (n>150 cells per condition). Red bars indicate the mean and error bars are SEM

4. For Fig 3, does USP50 deubiquitinate WRN? What is the mechanism that USP50 promote WRN to load to replication fork (should be confirmed by iPOND), bind to Fen1 and 9-1-1?

As discussed above, USP50 lacking all potential catalytic residues acts as WT and can support cell survival and suppress spontaneous 53BP1 foci, indicating that catalytic activity is not related to these phenomena (Supplemental Figure 3 I-K, O). Nevertheless, following up on the reviewer's comment, we examined WRN turnover. Intriguingly, there is an impact of USP50 on WRN turnover rates in HU-treated cells; however, as this is rescued by the IR mutant -we can conclude that the impact on WRN turnover is unlikely to relate to the impact on replication kinetics (which are disrupted by the mutant). This data is now included (Supplemental Figure 5B).

We performed several experiments to test the impact of USP50 on WRN:FEN1 further. We found a reduced ability of WRN to immunoprecipitate FEN1 in USP50-depleted cells (Figure 3G). While we could not confirm WRN interaction with nascent DNA using IPOND, we were able to investigate FEN1, finding that it shows reduced interaction with nascent DNA following USP50 loss and HU-treatment (Figure 3I). We conclude that USP50 promotes WRN-FEN1 presence at stalled forks. We have also confirmed that DNA2, RECQL4 and RECQL5 show increased association at or near nascent DNA, particularly as forks recover from HU (Figure 4 A & Bm Figure 5G-I).

5. Another big gas here is that why and how WRN/FEN1 switch to RECQL/DNA2 at the fork when USP50 is depleted? What is the role of USP50 in this process?

Given the impact of USP50 loss on the ability of FEN1 to associate with nascent DNA, we tested the possibility that FEN1 reduction might be critical to the switch. In support of this idea, we find that poor fork restart in cells depleted of FEN1 also show dependency on DNA2 and RECQL4/5 (Figure 5F). This observation is consistent with the notion that the reduced FEN1 observed on USP50 depletion contributes to the switch.

6. Most of the knockdown strategies utilizing in the manuscript are only one target, either siRNA or shRNA, another si/shRNA should be used to exclude the off target effect.

To address this, we used three separate siRNA sequences, tested targeting of USP50 in another human cell line and the targeting of murine USP50 in a mouse cell line, using spontaneous 53BP1-foci as a read-out. In all cases (four separate means), targeting USP50 resulted in increased 53BP1 foci (Supplemental Figure 3 L-N).

7. In MSI-H colorectal cell lines, WRN loss results in severe genome integrity defects and decreased viability. However, in Fig 3H HCT116 cells, depleted of USP50 doesn't affect cell viability but WRN depletion show dramatic effect? Is that suggest USP50 doesn't regulates WRN function here?

Yes, as discussed in the original manuscript – we conclude that USP50 does not contribute to WRN's ability to process the TA-rich stem loops associated with MSI-H cells. This was made clear in the original manuscript. We hope that in the rephrasing, it is clearer in the revised version.

References for the reviewer -please see the end of this document.

Reviewer #2 (Remarks to the Author):

In this manuscript, Morris and co-workers describe a protective role for USP50, a poorly understood and catalytically-inactive deubiquitinase (DUB), at DNA replication forks (RFs) required for suppressing DSB formation at specific genomic loci. As such the work is original and of considerable interest to the DNA repair and ubiquitin signaling community. Overall, most of their data is consistent with the view that USP50 localizes to chromatin in a ubiquitin(Ub)-binding-dependent manner (substrate(s) unknown) to promote the preferential use of DNA helicase:nuclease complexes (WRN:FEN1 over RECQL4/5:DNA2) at unperturbed and stressed RFs. However, there are quite some inconsistent (or missing) data and technical issues in the manuscript requiring major revisions that must be addressed before I can recommend it for publication.

1) Why did the authors choose to work on USP50 in the first place? It is stated (lane 99) that “Two previous RNA interference screens have highlighted USP50 as potentially important to replication.” In the first study, the Morris lab perform an siRNA screen to identify DUBs implicated in the clearance of Ub conjugates after release from hydroxyurea (HU) (Butler et al., 2012). In this paper, however, there is no screen data included. Therefore, I would ask the authors to provide this data. The second study provides at least the siRNA screen data showing that USP50 depletion does results in reduced cellular survival upon HU treatment.

We chose USP50 as the Yuan *et al.* 2014 study found that USP50 shRNA reduced cellular survival on HU treatment; these authors made no further follow-up of USP50. The screen from our work in Figure 1a of Butler et al. 2012, as the reviewers say, is a siRNA screen of FK2-conjugates (not reproduced for copyright reasons). USP50 siRNA-treated cells showed the lowest FK2 levels of any siRNA tested. However, our original screen was not controlled for cell numbers and is thus only tentatively pointed to a role for any DUB. Given the design of our previous screen, we have removed the reference to it in the current manuscript, pointing instead only to the study by Yuan et al.

2) In Figure 1, there is quite some additional experimental data required to support USP50 binding to Ub conjugates via a conserved pocket (I141) is promoting its localization to stalled RFs:

- a) To show that USP50 binding to Ub conjugates supports USP50 chromatin association, they should include Ub-I44A mutant (Figure 1B) and an E1 inhibitor (Figures 1B and 1D)
- b) Figure 1C is redundant with and less convincing than Figure 1F, and could be easily removed.
- c) Data shown in Figure 1D raises many questions: Is USP50 itself conjugated with ubiquitin, (i.e., polyubiquitinated) and perhaps degraded on chromatin? Why did VCPi not work in the same manner as MG132 (I suggest prolonging treatment with VCPi to 6 hours according to Anderson, Cancer Cell 2015)? Finally, also this experiment should be conducted with the I141R mutant and with an E1 inhibitor to substantiate the conclusions.
- d) The experimental set up of Figure 1E (EdU treatment was for 24 hours) is quite unusual. In principle a co-localization of USP50 with PCNA foci would be more informative. There is no concluding statement in the text for this data. I would suggest removing but include in Figure 1G (EdU pulse) samples without HU treatment to unambiguously illustrate “USP50 enrichment at stalled forks”.
- e) Figure 1F highlights an important finding, yet the western blots are extremely small and the signal for USP50 quite weak

We have performed several experiments to test this question further. We attempted to make bacterially expressed USP50 in order to assess direct Ub binding. USP50 expression was very low in bacteria so we turned to the expertise of a lab that makes bacteria- expressed proteins regularly (we do not), and they failed to make it entirely. We therefore performed very large grows and were eventually able to purify sufficient protein (the IR mutant was not possible) to test. This approach allowed us to assess whether purified MBP-fused USP50 could bind purified Ub conjugates and we found that it could (Supplemental Figure 2B).

We undertook the suggested VCP experiment. Longer exposure (6 hr) increased the degree of Ub on chromatin, but interestingly had no impact on USP50-chromatin association (Figure R2 for the reviewer). The Ub requirements for proteasome Vs p97/VCP interaction differ considerably, with the later requiring chains >5xUb^{6,7}. It is perhaps relevant that p97/VCP suppression has very little impact on proteins associated with nascent DNA⁸. These findings suggest some discrimination, we speculate that the Ub that USP50 is associated with are not long enough to drive p97 interaction.

We agree that USP50 is also Ub modified and turned over in a likely UPS-dependent manner (as the higher molecular-weight bands after MG132-treatment show, Figure 1D long exposure). We have been aware that UPS regulation of USP50 levels may confound the interpretation of our data and have tried to control for it, or reduce its impact, where possible. For example, following the reviewers suggestion we tested the E1 inhibitor (TAK243), finding it has a dramatic impact on USP50 protein levels (resulting in very large increases) (Figure R3 for the reviewer). These data are more evidence that USP50 is regulated through the Ub-system but also mean we cannot say anything about the increase of USP50 on chromatin under conditions of E1 inhibition (or prolonged MG132 where the same phenomena occurs), since our experiments are confounded by the hugely increased USP50 levels. In contrast, short-term MG132 treatment had marginal impact on USP50 levels, allowing us to compare its impact on USP50 chromatin accumulation independent of USP50 expression levels (Figure 1E).

We appreciate the suggestion of testing I44A-Ub expression on the ability of USP50 to associate with forks – however, the mutation is reported to have a profound impact on conjugation⁹ and is likely to have severe pleiotropic effects, so we were cautious of this approach. We have performed a series of depletion experiments removing some of the E3 ligases known to be important in replication dynamics (see response to reviewer 3). However, as yet, none of these suppressed USP50-accumulation on chromatin. Thus, instead we tested the impact of Ub over-expression, and prior MG132-treatment and used very short EdU incubation (5 min) to be sure to be assessing -as much as possible- proximity to nascent DNA. Suppressing the proteasome increases Ub in current conjugates and also rapidly inhibits free Ub levels, thereby suppressing new conjugate formation¹⁰. Consistent with Ub-regulated proximity of USP50 to nascent DNA, the signal between FLAG-USP50 and EdU was increased following Ub over-expression and reduced when MG132 was added prior to EdU (Figure 1F). We also tested the impact of Ub over-expression on the ability of the mutant to co-locate with nascent DNA, and while it has less over-all proximity than WT, we do see an increase on Ub over-expression – thus we cannot conclude that the ile-141 face is solely responsible for recruitment or the Ub-reading of USP50 that locates it near to nascent DNA. Thus we agree that the statement that USP50 recruitment to forks directly depends on a Ub:Ile-141 face interaction, cannot be substantiated by the current evidence. Importantly in the

Figure R2. Impact on 6 hours VCP inhibition. Cells expressing FLAG-USP50, or not, were exposed to 6h VCPi (2.5 μ M) and then 3h HU (5 mM) before the processing for the chromatin fraction and western blotting was performed, probing for FLAG-USP50, Ubiquitin (FK2) and Histone H3.

Figure R3. TAK243 treatment increases USP50 levels. HeLa cells expressing FLAG-USP50 were left untreated or treated with 10 μ M TAK243 for 6 hours. Cells were then lysed and western blotting was performed, probing for FLAG-USP50 and Histone H3.

revised manuscript, we do not make that claim and have been careful not to overstate and to be clear about where the knowledge gaps remain.

The experimental set up of Figure 1E (EdU treatment was for 24 hours) is quite unusual.

Indeed, our use of 24-hour EdU is not commonplace. This long 24-hour incubation is not, as suggested, used to look at proximity to sites of replication but to address interaction with DNA as a whole (the 24-hour incubation allows full DNA labelling). Figure 1E EdU treatment was for 24-hours was undertaken to enable a direct comparison with the short-term, more widely used, 15-minute EdU treatment. We had not communicated the point of these experiments well, so we rearranged (to put side-by-side) and similarly reordered the text and more clearly marked the purpose of the approach. PCNA association would assess interaction with replication factories, similar to the short-term EdU association that we undertook. We have also extended the concluding text statement.

The redundant blot has been removed and “Figure 1F (now Figure 1C) western blots increased in size to show USP50 is at a good level.

3) In Figure 2, the authors design a stable cell system where the shRNA against endogenous USP50 as well as the expression of FLAG-tagged USP50 can be induced. Even though it may be true that HeLa cells express very low levels of USP50, I would still highly recommend confirming USP50 depletion at least at the mRNA level using RT-PCR. In fact, it seems as if expression of the USP50-I141R mutant has a dominant-negative effect (in absence of shUSP50 expression), suggesting that the phenotypes seen with shUSP50 expression could be caused by an off-target effect. One might wonder whether depletion of a factor that is expressed at undetectable levels can produce any specific phenotype? Therefore, it is crucial to confirm at least a few results with a different RNAi-targeting sequence (i.e., with a second shRNA against USP50)

We tested three different sequences in HeLa cells, and tested USP50 siRNA in MCF7 cells, and mouse USP50 siRNA in NIH3T3 cells for the ability to suppress spontaneous 53BP1 foci (Supplemental Figure 3L-N). In each case, reagents targeting USP50 resulted in increased 53BP1 foci. These data strongly argue against an off-target effect since the same phenotype is invoked by multiple different agents targeting USP50 and does so in human and mouse cells.

4) In Figure 2D, and again in Figure 4F, attempting to provide experimental evidence for increased ssDNA formation in response to HU treatment in USP50-depleted cells, the authors show very poor and unconvincing western blot data for phosphorylated RPA2 (as a ssDNA marker). These blots have either to be significantly improved (total RPA2 is also missing in Figure 4F) or an alternative readout should be used. For instance, others have employed immunofluorescence microscopy analysis of BrdU staining under non-denaturing conditions after HU treatment (e.g., Leung et al Cell Reports 2023).

Multiple pRPA blots have been performed and quantified (Figure 2D), in addition, we have used a native BrdU-detection method to measure ssDNA tracts¹¹, which we show in Figure 2E and Figure 5D.

5) Increased 53BP1 foci formation and reduced survival following replication stress (induced by HU and PDS) could also indicate a defect in HR-mediated fork restart in cells depleted of USP50. It could be informative to examine whether USP50 depletion impairs HR using classical DR-GFP reporter cells.

We have undertaken these assays and find the results do not support a significant role for USP50 in canonical HR, or NHEJ (Figure R4 for the reviewer below).

Figure R4. Impact of USP50 depletion on DNA double-strand break repair.

A) Representation of NHEJ GFP assay. B) Representation of HR GFP assay. C) Percent of GFP and RFP positive cells as a proportion of NTC cells. USP50 siRNA did not cause a large decrease in the cells ability to repair by NHEJ. D) Percent of GFP and RFP positive cells as a proportion of the NTC control siRNA treated cells. Experiments were carried out in triplicate (n=3).

6) Data in Figure 3D suggesting USP50-dependent interaction between WRN and FEN1 (Figure 3D) based on in situ proximity assays are not convincing (no representative images are shown) and must be corroborated with co-IP data.

We have examined WRN:FEN1 interaction by immunoprecipitation, noting a slight reduction in interaction following USP50 depletion (Figure 3G), consistent with the WRN:FEN1 PLA data. Moreover, we find a reduction of FEN1 in IPOND samples in USP50-depleted cells after HU (Figure 3I).

7) Figures 3E-G highlight the functional interplay between USP50 and WRN:FEN1 at both unperturbed and HU-challenged RFs. However, they are all based on different experimental readouts (53BP1 foci after HU, ongoing replication using different DNA fiber labelling schemes). To corroborate these important mechanistic findings, I would strongly recommend using for all conditions (overexpression of WRN and FEN1 variants) the same readouts without HU (e.g., 1st label termination or fork asymmetry) and with HU (% stalled forks (3A) and 53BP1 foci (3E)).

We have performed new experiments for each, confirming our findings at each stage (Figure 3D, E, Figure 4C D, Supplemental Figure 5 E & D, Figures 5 A, B C and E).

8) Figure 3I indicates a role for USP50 in telomere maintenance. Is this function dependent on Ub-binding (use IR mutant) and on USP50-mediated WRN-FEN1 activity at RFs (use siWRN and siFEN1 (as in 3A) and/or WRN WT and mutant, FEN1 WT and mutant expression constructs (shown in Figure 3F and 3G))?

We performed a new set of telomere examination experiments, indeed finding that complementation with WT-USP50, but not the IR mutant, can suppress telomere loss (Supplemental Figure 5L). As the role of WRN and FEN1 in telomere stability has been extensively studied^{12, 13, 14, 15, 16} – we instead asked the next question, whether the suppression of DNA2 or RECQL4/5, that rescue features of USP50-depleted cell fork kinetics (and we now show also rescue fork restart in FEN1 depleted cells, Figure 5F) would impact the telomere loss. We found that, indeed, suppression of DNA2 was able to prevent telomere loss in USP50-depleted cells (Figure 4H).

9) The increase in DNA2 foci numbers following USP50 depletion in HU-treated cells is very minimal and barely visible (Figure 4A, average foci number per cell increases from 12 to 15) and could simply be due to a

slight increase in DNA2 protein levels (Suppl Figure 3I). I suggest checking DNA2 levels on chromatin after HU in control- and USP50-depleted cells by western blotting to substantiate the microscopy data. Similarly, could there be an increase of RECL4/5 levels at chromatin after HU upon USP50 depletion (like what was observed for WRN (Figure 3B)?

We have checked DNA2 levels (and indeed REQL4 and RECQL5 levels) which are unaltered by USP50 loss (Supplemental Figure 6A, 7C and 7D). We have performed a further series of experiments to assess the association of DNA2, RECQL4 and RECQL5 at nascent DNA with or without USP50 depletion on HU treatment and on release from HU (Figures 4B and Figure 5G-I). These assays use PLA and immunofluorescence and show, as the reviewer suspected, an increased association with nascent DNA in cells lacking USP50.

10) It would be quite revealing to test whether USP50 'directly' associates with any of the nuclease helicase factors in an Ub-dependent manner? Along these lines, perhaps USP50 regulates the resection of certain specific DSBs via controlling the access of WRN, thereby rejecting DNA2, two proteins implicated in DSB resection.

Indeed, our data points to this sort of mechanism. In the manuscript we show that DNA2 and RECQL4/5 helicases are – as the reviewer speculated- responsible for the increased ssDNA observed in USP50 cells (Figure 5D), and they are also responsible for poor fork progression in USP50 (Figure 4D, Figure 5 A & B) or FEN1 (Figure 5F) depleted cells. As to a direct mechanism we do indeed find USP50 and FEN1 co-precipitate (Figure R5 for the reviewer). We consider that the underpinnings of this require more investigation (confirmation that this is Ub-dependent identification of Ub-sites, identification the ligase etc) and don't feel it is sufficient to report in the current manuscript.

Figure R5. Interaction between USP50 and FEN1 on HU. HeLa cells were co-treated with +/- 10 μ M MG132, +/- 5mM HU, +/- shUSP50 and +/- siFEN1 before harvesting in nuclear lysis buffer and incubated with FLAG-agarose beads overnight. The beads were spun down and samples eluted off the beads before western blotting was performed. Membranes were probed for FLAG-USP50 and endogenous FEN1.

Minor comments:

- I assume that in Figure 2E, HU treatment is mis-labeled as EdU+. **Relabelled**
- Include HU treatment in the Figure legends for panels 2F, 2G and 3E. **Included.**

References for this reviewer-please see the end of this document.

Reviewer #3 (Remarks to the Author):

In this manuscript, the authors employed human cell lines and various cell biological and biochemical techniques to investigate the role of USP50 in DNA replication. They demonstrated that the inactivation of USP50 affects cell survival in response to HU-induced replication stress, yet the mechanistic insight into its

role in DNA replication remains elusive. Based on their experiments, predominantly utilizing the DNA fiber assay, the authors propose a model in which USP50 binds to ubiquitinated substrate X on chromatin at the sites of DNA replication forks/stalled forks, thereby stimulating the recruitment of the WRN and FEN1 complex, which coordinates unperturbed DNA synthesis and fork restart. In contrast, the absence of USP50 results in reduced WRN:FEN1 complex presence at stalled replication fork/chromatin sites, allowing for an increased presence of DNA2, RecQL4/5, whose nuclease and helicase activities negatively affect the stability and restart of DNA replication forks, particularly at some GC-rich regions.

This is a well-structured manuscript; however, it requires additional confirmation of a few key conclusions for publication in a reputable scientific journal. Two critical points need addressing:

First, the authors must provide direct evidence of USP50 inactivation leading to a reduction in the levels of WRN and FEN1 proteins at the replication forks, simultaneously with an increase in the levels of DNA2 and RecQL4/5. While Figure 4A shows increased DNA2 foci in USP50 cells, this effect should be substantiated through iPOND technology or by examining isolated chromatin from S-phase cells to elegantly demonstrate these changes.

We performed IPOND. FEN1 and PCNA were detected in control IPOND samples, and we found that FEN1 was reduced on nascent DNA in USP50-depleted cells following HU, compared to control cell nascent DNA after HU-treatment (Figure 3I). These data provide further evidence of USP50 regulation of FEN1 on nascent DNA. However, as previously described^{1, 17} we were unable to obtain a signal for WRN, or RECQL4/5 in these assays utilising specific antibodies. We tested variants of the IPOND protocol designed to improve the detection of larger proteins (RIPA buffer and diluted SDS¹ and native²) as well as testing several of our own changes. However, no condition was satisfactory for these proteins.

To further test the association, we used the analogous methodology of EdU incorporation followed by detection of the recently incorporated DNA -and antibodies to DNA2, RECQL4 and RECQL5 we tested proximity to nascent DNA on HU, and after its wash-out (Figures 4B and Figure 5F, G and H). These assays show an increased association of the three proteins with nascent DNA in cells lacking USP50.

Indeed the same PLA approach, using short-term EdU incorporation, has been used in recent investigations by others, for example, to investigate MRE11 and EXO1 interaction with nascent DNA at the stalled replication forks, Nusawardhana, *et al.* Nucleic acid research 2024 ⁴ and to assess the localization of the newly identified reversal factor TFIPII with nascent DNA at forks, Chen *et al* Nature Comms 2024, ⁵. To test it further we also used very short EdU incorporation times (5 mins) and also performed a thymidine chase with excess thymidine for 10 minutes before HU treatment (Figure 1F). This experiment confirmed the requirement on nascent DNA to induce the signal. We have been careful not to overclaim and state our data shows USP50 “at or near” nascent DNA.

Secondly, the authors should validate their model by assessing its impact on cell survival. The diminished survival of USP50-deficient cells in response to HU and Pyr, as observed in Figure 2H and I, respectively, should be rescued when DNA2 or RecQL4/5 is co-inactivated or co-depleted.

We tested the impact on cell survival. Remarkably, DNA2 inhibition, but not depletion of RECQL4 and improved the viability of USP50-depleted cells to HU (Figure 3I). There was also a non-significant improvement of cell survival after exposure to pyridostatin following co-depletion of RECQL4/5, but not with DNA2 (Figure 5J). We thank the reviewer for the suggestion.

Specific comments:

1. Figure 1B: The authors should also analyze the total ubiquitylation signal on the membranes. As part of the quantification, we did analyse the total ubiquitin signal for the inputs and pulldown and used this to normalise the results to ensure any differences in transfection were taken into account. The results are provided in the source data files.

2. Figure 1D: The 5 μ M/4hr MG132 treatment should deplete the nuclear pool of ubiquitin. If their model is correct, USP50 recruitment to chromatin should not occur in the absence of ubiquitin in the nucleus. The results in this figure suggest that USP50 chromatin removal depends on the proteasome.

Indeed, MG132-treatment acts to starve the cells of free Ub, preventing new Ub addition¹⁰. At the same time, Ub conjugates that are already present and processed through the proteasome are stabilised, and Ub is not lost from the nucleus (e.g.¹⁸); it is the latter phenomenon that appear to be present here on chromatin where the Ub-conjugates are greatly increased by MG132 treatment (Figure 1D). We also agree with the reviewer that USP50 is itself a target for proteasomal turnover, as can be seen on the blot.

As suppressing the proteasome also rapidly inhibits free Ub levels, thereby suppressing new conjugate formation¹⁸ we tested the impact of MG132-treatment prior to EdU incubation (5 min) to be sure to be assessing -as much as possible- proximity to nascent DNA. Consistent with Ub-regulated proximity of USP50 to nascent DNA, the signal between FLAG-USP50 and EdU was reduced when MG132 was added prior to EdU (Figure 1F).

3. Figure 2D: The authors should also demonstrate the restoration of ssDNA formation through the overexpression of USP50-wt but not USP50-IR. This experiment has been undertaken, Figure 2E.

4. Figure 3B: This is a crucial experiment for their model. However, the WRN signal on the Western blot is weak. The authors need to improve the quality of this experiment and determine whether this phenotype is restored with the overexpression of USP50wt and USP50-IR. As suggested earlier in this report, the iPOND assay is the best method to demonstrate that WRN is reduced at stalled DNA replication forks in USP50-depleted cells.

We spent much time and effort trying to observe WRN via IPOND, using several variants of the technique that claim to improve the detection of large proteins such as WRN (such as RIPA buffer and diluted SDS¹ and native² without success (a few examples of our experiments are at the bottom of this response). Fortunately, we were able to observe FEN1 using IPOND and confirmed that FEN1 was reduced on nascent DNA in USP50-depleted cells following HU, compared to control cell nascent DNA after HU-treatment (Figure 3I). Thus our evidence of WRN-related phenotype is restricted to epistasis (Figure 3A), the reduced chromatin recruitment of WRN on HU (Figure 3B), reduced proximity to HUS1 (Figure 3C), the ability of WT-USP50, but not IR to encourage WRN-FEN1 proximity (Figure 3H), the ability of exogenous WRN to rescue the need for USP50 (Figure 3D & E and Supplemental figure 5D & E), and the failure of FEN1- bearing a mutation that disrupts WRN interaction to rescue all USP50 phenotypes (Figure 3K-M). For these reasons, we have altered the discussion text to highlight the weaknesses in our evidence for WRN modulation and the relative strength in the FEN1 observations.

5. Figure 4A and K: The quality of the foci images should be improved. Undertaken – and further assays done (Figure 4B).

6. As previously suggested, the authors need to demonstrate an increased amount of DNA2 and RecQL4/5 protein at the fork (using iPOND) or replicative chromatin (in S-phase) in USP50-depleted cells. They should then show the restoration of this phenotype with the overexpression of USP50 wt, but not USP50-IR.

We would very much like to present IPOND data for these proteins, but despite tremendous efforts from several members of the group, attempting multiple variants of the assay, we have been unable to observe them by IPOND. Instead, we used an analogous method (iPOND addresses proteins cross-linked to EdU-EdU-incorporated DNA and associated proteins; PLA addresses whether proteins are within 40 nm of EdU-EdU-incorporated DNA). The PLA analysis shows altered DNA2, RECQL4 and RECQL5 in proximity to EdU-label in USP50 depleted cells (Figures 4B and Figure 5G- I). As for WRN we have highlighted the limitations of our approach in the discussion. We have been careful not to overclaim and state our data shows these proteins “at or near” nascent DNA. As a reminder several authors have used the PLA approach in similar experiments for example, to investigate MRE11 and EXO1 interaction with nascent DNA at the stalled replication forks, Nusawardhana, *et al.* Nucleic acid research 2024 ⁴ and to assess the localization of the newly identified reversal factor TFIP11 with nascent DNA at forks, Chen *et al* Nature Comms 2024, ⁵.

7. The authors should use a CFA assay as in Figure 2H or I to measure cell survival in USP50-depleted cells and demonstrate rescue through co-depletion of DNA2 or RecQL4/5 and expression of WRN-wt, but not WRN-enzymatically inactive variants.

This has been undertaken DNA2 (Figure 4I), RECQL4/5 (Figure 5J), and WRN variants have been assessed for their ability to suppress ongoing fork stalling (Figure 3D), fork asymmetry (Supplemental Figure 5E) fork recovery from HU (Supplemental Figure 5D) and the ability to suppress 53BP1 foci (Supplemental Figure 5E).

Minor points and suggestions:

1. Instead of using a CldU-HU-IdU assay to measure stalled forks, consider using this assay to measure fork restart. We considered presenting the data this way round. It's the same data whichever way (ability to start, or more forks stalled).

2. It is not clear how reference 41 (Butler et al., 2012) highlights the role of USP50 in DNA replication. The authors should clarify whether siUSP50 increases or decreases the total Ub/FK2-signal based on the original paper. The Butler reference is now removed. In our original screen, Butler *et al.*, no account was made for cell numbers (cell death on HU would have reduced the FK2 signal).

3. Investigate well-known E3-ubiquitin ligases involved in DNA replication and replication stress response, such as RNF8, RNF168, BRCA1/BARD1, and Cullin-RING ligases, to support the Ub-based model. For cullin-RING ligases, utilizing a NEDD8 inhibitor may be sufficient.

A selection of some of the usual suspects are shown in Figure R6 for the reviewer. In addition, we also tested RNF20/40 and USP22 siRNAs and the inhibition of RING1B, using the inhibitor PRT4165 in the presence of cycloheximide. As yet none of these approaches suppressed USP50-chromatin enrichment (hence the testing of prior MG132 treatment to suppress new Ub conjugates). A full screen is needed to identify the relevant ligase(s).

All 3h HU @ 5 mM, 72h knockdowns with 25 nM siRNA

Figure R6. Selection of E3-ligase siRNA treatments do not suppress USP50 - chromatin association. HeLa cells expressing FLAG-USP50 treated for 72 hours with the siRNAs indicated before lysing for whole cell or chromatin fraction and blotting for USP50 and histone H3.

4. Supplementary Figure 4 should be Supplementary Figure 1. Re-ordered

References for this reviewer-please see the very end of this document (below).

For all reviewers: A subset of our attempts to resolve WRN by IPOND:

Figure R7. iPOND was carried out following the protocol as described in materials & methods. A whole cell extract sample is also included and was lysed prior to the iPOND protocol in order to determine the effect of the process on WRN levels. As shown, we are able to detect WRN in the WCE but are losing material/unable to detect in the iPOND.

Figure R8. HeLa cells were lysed and subjected to varying concentrations of PFA for crosslinking to optimise conditions. WRN levels indicate crosslinking and sonication is working (we see a shifted band if we don't decrosslink), and they don't appear to be the issue preventing WRN detection.

Figure R9. HeLa cells were treated with a variety of different permeabilisation conditions in an attempt to retain the WRN signal seen under untreated conditions. WRN levels indicate that permeabilisation seems to be the issue causing the loss of WRN. This cannot be overcome by altering the permeabilisation conditions.

Figure R10. HeLa cells were subject to a further subset of permeabilization conditions, confirming this is the cause for our loss of WRN signal, and this cannot be overcome by drastically changing conditions with milder permeabilisation or stronger crosslinking. HeLa cells were also plated for IF which indicates EdU-Click can penetrate cells without permeabilisation, after PFA crosslinking.

Figure R11. iPOND was carried out following the protocol as described in materials & methods and shown in **Figure R7**, but without the permeabilisation step. WRN and DNA2 blots show we are unable to detect large molecular weight proteins in the inputs or iPOND samples.

References for reviewers.

1. Wang Q, *et al.* Modified iPOND revealed the role of mutant p53 in promoting helicase function and telomere maintenance. *Aging (Albany NY)* **15**, 10767-10784 (2023).
2. Wiest NE, Tomkinson AE. Optimization of Native and Formaldehyde iPOND Techniques for Use in Suspension Cells. *Methods Enzymol* **591**, 1-32 (2017).
3. Alam MS. Proximity Ligation Assay (PLA). *Curr Protoc Immunol* **123**, e58 (2018).
4. Nusawardhana A, Pale LM, Nicolae CM, Moldovan GL. USP1-dependent nucleolytic expansion of PRIMPOL-generated nascent DNA strand discontinuities during replication stress. *Nucleic Acids Res* **52**, 2340-2354 (2024).
5. Chen J, *et al.* TFIP11 promotes replication fork reversal to preserve genome stability. *Nat Commun* **15**, 1262 (2024).
6. Saeki Y. Ubiquitin recognition by the proteasome. *J Biochem* **161**, 113-124 (2017).
7. Fujisawa R, Polo Rivera C, Labib KPM. Multiple UBX proteins reduce the ubiquitin threshold of the mammalian p97-UFD1-NPL4 unfoldase. *Elife* **11**, (2022).
8. Rivard RS, *et al.* Improved detection of DNA replication fork-associated proteins. *Cell Rep* **43**, 114178 (2024).
9. Saha A, Lewis S, Kleiger G, Kuhlman B, Deshaies RJ. Essential role for ubiquitin-ubiquitin-conjugating enzyme interaction in ubiquitin discharge from Cdc34 to substrate. *Molecular Cell* **42**, 75-83 (2011).
10. Xu Q, Farah M, Webster JM, Wojcikiewicz RJ. Bortezomib rapidly suppresses ubiquitin thiolesterification to ubiquitin-conjugating enzymes and inhibits ubiquitination of histones and type I inositol 1,4,5-trisphosphate receptor. *Mol Cancer Ther* **3**, 1263-1269 (2004).
11. Altieri A, Dell'Aquila M, Pentimalli F, Giordano A, Alfano L. SMART (Single Molecule Analysis of Resection Tracks) Technique for Assessing DNA end-Resection in Response to DNA Damage. *Bio Protoc* **10**, e3701 (2020).
12. Crabbe L, Verdun RE, Haggblom CI, Karlseder J. Defective telomere lagging strand synthesis in cells lacking WRN helicase activity. *Science* **306**, 1951-1953 (2004).
13. Saharia A, *et al.* Flap endonuclease 1 contributes to telomere stability. *Curr Biol* **18**, 496-500 (2008).
14. Damerla RR, Knickelbein KE, Strutt S, Liu FJ, Wang H, Opresko PL. Werner syndrome protein suppresses the formation of large deletions during the replication of human telomeric sequences. *Cell Cycle* **11**, 3036-3044 (2012).
15. Saharia A, Stewart SA. FEN1 contributes to telomere stability in ALT-positive tumor cells. *Oncogene* **28**, 1162-1167 (2009).
16. Saharia A, Teasley DC, Duxin JP, Dao B, Chiappinelli KB, Stewart SA. FEN1 ensures telomere stability by facilitating replication fork re-initiation. *J Biol Chem* **285**, 27057-27066 (2010).
17. Kehrl K, Phelps M, Lazarchuk P, Chen E, Monnat R, Jr., Sidorova JM. Class I Histone Deacetylase HDAC1 and WRN RECQ Helicase Contribute Additively to Protect Replication Forks upon Hydroxyurea-induced Arrest. *J Biol Chem* **291**, 24487-24503 (2016).
18. Stack JH, Whitney M, Rodems SM, Pollok BA. A ubiquitin-based tagging system for controlled modulation of protein stability. *Nat Biotechnol* **18**, 1298-1302 (2000).

REVIEWERS' COMMENTS

Reviewer #1 (Remarks to the Author):

My questions are addressed.

Reviewer #2 (Remarks to the Author):

In their point-by-point letter, the authors have made a great effort to address all concerns previously raised and provided a significant number of new and conclusive experimental data. Overall, I believe the manuscript has substantially improved and now meets the required quality for publication.

I do however think that the Model in Figure 6 and especially the very minimal figure legend requires quite some editing (there are quite some typos) to make them coherent with the data and more comprehensible for the reader.

Therefore, after revising Figure 6, my recommendation is to publish the MS in its present form.

Reviewer #3 (Remarks to the Author):

The authors have addressed the majority of my criticisms, and I support the acceptance of this manuscript after minor stylistic changes to the main text.

Specifically,

Lane 119: The sentence: "Ub-conjugates can be directed for degradation by the proteasomal or unwound by the p97 AAA+ ATPase segregase" should be: Ub-conjugates can be directed for degradation by the proteasome or unwound by the p97 AAA+ ATPase segregase.

In line 122, the authors mention the p97 inhibitor CB-5083 and ref.45. In the indicated reference, there is nothing about p97 inhibitors, and the review article reference on p97 is from 2014. There has been much progress made on the role of p97 in chromatin since 2014. Therefore, the recently published review article (PMID: 36640759) better explains the role of p97 and its inhibitors, including CB-5083. This reference should be cited as well."

Response to reviewers

REVIEWERS' COMMENTS

Reviewer #1 (Remarks to the Author):

My questions are addressed.

Many thanks

Reviewer #2 (Remarks to the Author):

In their point-by-point letter, the authors have made a great effort to address all concerns previously raised and provided a significant number of new and conclusive experimental data. Overall, I believe the manuscript has substantially improved and now meets the required quality for publication.

I do however think that the Model in Figure 6 and especially the very minimal figure legend requires quite some editing (there are quite some typos) to make them coherent with the data and more comprehensible for the reader.

Therefore, after revising Figure 6, my recommendation is to publish the MS in its present form.

Many thanks -we have altered the figure and legend and hope these changes make it more comprehensible.

Reviewer #3 (Remarks to the Author):

The authors have addressed the majority of my criticisms, and I support the acceptance of this manuscript after minor stylistic changes to the main text.

Specifically,

Lane 119: The sentence: "Ub-conjugates can be directed for degradation by the proteasomal or unwound by the p97 AAA+ ATPase segregase" should be: Ub-conjugates can be directed for degradation by the proteasome or unwound by the p97 AAA+ ATPase segregase.

Many thanks – corrected.

In line 122, the authors mention the p97 inhibitor CB-5083 and ref.45. In the indicated reference, there is nothing about p97 inhibitors, and the review article reference on p97 is from 2014. There has been much progress made on the role of p97 in chromatin since 2014. Therefore, the recently published review article (PMID: 36640759) better explains the role of p97 and its inhibitors, including CB-5083. This reference should be cited as well."

Many thanks – this is indeed a much better reference (and a nice review). Now included.